



# Global Ozone Monitoring Experiment-2 (GOME-2) Daily and Monthly Level 3 Products of Atmospheric Trace Gas Columns

Ka Lok Chan[1,2], Pieter Valks[2], Klaus-Peter Heue[2,3], Ronny Lutz[2], Pascal Hedelt[2], Diego Loyola[2], Gaia Pinardi[4], Michel Van Roozendael[4], François Hendrick[4], Thomas Wagner[5], Vinod Kumar[5], Alkis Bais[6], Ankie Piters[7], Hitoshi Irie[8], Hisahiro Takashima[9,10], Yugo Kanaya[10], Yongjoo Choi[11], Kihong Park[12], Jihyo Chong[12,13], Alexander Cede[14], Udo Frieß[15], Andreas Richter[16], Jianzhong Ma[17], Nuria Benavent[18], Robert Holla[19], Oleg Postylyakov[20], Claudia Rivera Cárdenas[21], and Mark Wenig[22]

[1]Rutherford Appleton Laboratory Space, Harwell Oxford, United Kingdom
[2]Remote Sensing Technology Institute, German Aerospace Center (DLR), Oberpfaffenhofen, Germany
[3]Department of Aerospace and Geodesy, Technical University of Munich (TUM), Munich, Germany
[4]Royal Belgian Institute for Space Aeronomy (BIRA-IASB), Brussels, Belgium
[5]Max Planck Institute for Chemistry (MPIC), Mainz, Germany
[6]Laboratory of Atmospheric Physics, Aristotle University of Thessaloniki (AUTH), Thessaloniki, Greece
[7]Royal Netherlands Meteorological Institute (KNMI), De Bilt, Netherlands
[8]Center for Environmental Remote Sensing (CEReS), Chiba University, Chiba, Japan
[9]Faculty of Science, Fukuoka University, Fukuoka , Japan
[10]Research Institute for Global Change (RIGC), Japan Agency for Marine-Earth Science and Technology (JAMSTEC), Yokohama, Japan
[11]Department of Environmental Science, Hankuk University of Foreign Studies, Seoul, Korea
[12]School of Earth Sciences and Environmental Engineering, Gwangju Institute of Science and Technology, Gwangju, Korea
[13]Environmental Management Division, Yeongsan River Basin Environmental Office, Gwangju, Korea
[14]LuftBlick, Kreith, Austria
[15]Institute of Environmental Physics, University of Heidelberg, Heidelberg, Germany
[16]Institute of Environmental Physics, University of Bremen, Bremen, Germany
[17]Chinese Academy of Meteorology Science, China Meteorological Administration, Beijing, China
[18]Department of Atmospheric Chemistry and Climate, Institute of Physical Chemistry Rocasolano (CSIC), Madrid, Spain
[19]Deutscher Wetterdienst (DWD), Hohenpeißenberg, Germany
[20]A.M. Obukhov Institute of Atmospheric Physics (IAP), Russian Academy of Sciences (RAS), Moscow, Russia
[21]Instituto de Ciencias de la Atmósfera y Cambio Climático, Universidad Nacional Autónoma de México, Mexico City, Mexico
[22]Meteorological Institute (MIM), Ludwig-Maximilians-Universität München (LMU), Munich, Germany

**Correspondence:** Ka Lok Chan (ka.chan@stfc.ac.uk)

**Abstract.**

We introduce the new GOME-2 daily and monthly level 3 product of total column ozone ($O_3$), total and tropospheric column nitrogen dioxide ($NO_2$), total column water vapour, total column bromine oxide (BrO), total column formaldehyde (HCHO) and total column sulphur dioxide ($SO_2$). The GOME-2 level 3 products are aimed to provide easily translatable and user-

5 friendly data sets to the scientific community for scientific progress as well as satisfying public interest. The purpose of this paper is to present the theoretical basis as well as the verification and validation of the GOME-2 daily and monthly level 3 products.



The GOME-2 level 3 products are produced using the overlapping area weighting method. Details of the gridding algorithm are presented. The spatial resolution of the GOME-2 level 3 products is selected based on sensitivity study. The consistency of the resulting level 3 products among three GOME-2 sensors is investigated through time series of global averages, zonal averages, and bias. The accuracy of the products is validated by comparing to ground-based observations. The verification and validation results show that the GOME-2 level 3 products are consistent with the level 2 data. Small discrepancies are found among three GOME-2 sensors, which are mainly caused by the differences in instrument characteristic and level 2 processor. The comparison of GOME-2 level 3 products to ground-based observations in general shows very good agreement, indicating the products are consistent and fulfil the requirements to serve the scientific community and general public.

## 1    Introduction

Satellite remote sensing observations provide indispensable spatio-temporal information of atmospheric composition on a global scale. Various atmospheric trace gases can be retrieved from nadir viewing satellite spectroscopic observations in the ultraviolet (UV) and visible (Vis) spectral range. This type of observation has long been conducted since the Global Ozone Monitoring Experience (GOME) mission launched in 1995 (Burrows et al., 1999). Together with its successors, Global Ozone Monitoring Experience 2 (GOME-2; Callies et al. 2000), provided a global record of spectrally resolved earthshine radiance in the UV and Vis spectral range for more than 25 years.

The processing of satellite observation of trace gas column involves two major steps, (1) conversion of raw spectral channel counts (level 0 data) to geolocation and radiometric calibrated radiance and irradance data (level 1B data), and (2) retrieval of trace gas columns (level 2 data) from level 1B data. The retrieval of trace gas columns from level 1B data includes spectral retrieval of slant columns of trace gas and subsequently convert them to vertical columns. To ensure the accuracy and consistency of satellite observations, GOME-2 data are processed in stable operational environments within the framework of Satellite Application Facility on Atmospheric Composition Monitoring (AC SAF). The level 0 to level 1B conversion is processed by European Organisation for the Exploitation of Meteorological Satellites (EUMETSAT), while the level 1B to level 2 is processed by German Aerospace Center (DLR).

The GOME-2 level 2 data are orbital-swath scientific products. A level 2 data file contains observations within a single orbit. Trace gas columns are usually expressed in the satellite viewing geometry of reference using across-track and along-track position. In addition, the satellite measurement footprint is not in regular latitude-longitude grid and often multiple pixels overlapping at the edges of orbit within a day. Using this kind of scientific product requires very good knowledge of the satellite product, especially when averaging multiple measurements to generate daily or monthly maps. In order provide a user friendly satellite product, we have projected the GOME-2 level 2 data onto a regular latitude-longitude grid to generate operational level 3 products. The purpose of this document is to present the theoretical basis, verification and validation of the GOME-2 level 3 daily and monthly gridded products. These products includes global daily and monthly mean of total column ozone ($O_3$), total and tropospheric column nitrogen dioxide ($NO_2$), total column bromine oxide (BrO), total column water vapour ($H_2O$), total column formaldehyde (HCHO), and total column sulphur dioxide ($SO_2$).





The paper is organized as follows. Section 2.1 describes the GOME-2 instruments. A brief description of each GOME-2 level 2 trace gas product can be found in Section 2.2, while auxiliary data sets used for comparison are presented in Section 2.3. The gridding algorithm for level 2 to level 3 processing is presented in section 3.1. Section 3.2 shows the selection of best spatial resolution for GOME-2 level 3 data. The verification and validation methodology is presented in section 3.3. Section 4 presents the resulting daily and monthly level 3 products. Result for the verification and validation of GOME-2 level 3 data is presented in Section 5. Finally, Section 6 summarises our findings.

## 2 Instruments and data sets

The GOME-2 instruments and the operational level 2 products of each trace gas used for the generation of gridded level 3 data are described in this section. Ground-based measurements used to validate the GOME-2 level 3 products are also presented.

### 2.1 GOME-2 instruments

Global Ozone Monitoring Experiment 2 (GOME-2) are passive nadir viewing satellite borne spectrometers on board the European Organization for the Exploitation of Meteorological Satellites (EUMETSAT) MetOp series of satellites. The MetOp satellites orbit at an altitude of ~820 km on sun-synchronous orbits with 29 days (412 orbits) repeat cycle and a local equator overpass time of 09:30 LT (local time) on the descending node. MetOp-A, the first MetOp satellite, was launched on 19$^{th}$ October 2006. MetOp-B was launched 6 years later on 17$^{th}$ September 2012. The third MetOp satellite, MetOp-C, was launched on 7$^{th}$ November 2018. A more detailed introduction of the MetOp series of satellites can be found in Klaes et al. (2007).

The GOME-2 instruments are optical spectrometers equipped with scanning mirrors which enable across-track scanning in nadir and side ways viewing for polar coverage (Callies et al., 2000). Each GOME-2 instrument consists of four detectors covering a wavelength range of 240 - 790 nm with spectral resolution ranging from 0.26 nm to 0.51 nm. Additionally, two polarization components are measured with polarization measurement devices (PMDs) using 30 broadband channels covering the full spectral range at higher spatial resolution. The nominal spatial resolution of the instruments is 80 km (across-track) × 40 km (along-track) for the forward scan and the spatial resolution reduced to 240 km (across-track) × 40 km (along-track) for the backward scan. The scanning swath width of the GOME-2 instruments is about 1920 km. After the GOME-2 instrument on board the MetOp-B satellite (refers as GOME-2B from hereafter) went in tandem operation with MetOp-A in July 2013, the across-track spatial resolution of the GOME-2 instrument on board the MetOp-A satellite (refers as GOME-2A from hereafter) was doubled to 40 km with the spatial coverage of a swath reduced to 960 km. The spatial resolution and coverage of GOME-2B remains unchanged. A more detailed description of the GOME-2 instruments can be found in Munro et al. (2016). Table 1 summarized the major characteristics of all three GOME-2 instruments.

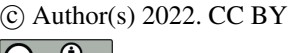



**Table 1.** Summary of the GOME-2 instrument characteristics.

| Sensor | GOME-2A | GOME-2B | GOME-2C |
|---|---|---|---|
| Operational Period | Jan 2007 - Nov 2021 | Dec 2012 - Present | Jan 2019 - Present |
| Spectral Range | 240 - 790 nm | 240 - 790 nm | 240 - 790 nm |
| Ground Pixel Resolution | 80 km $\times$ 40 km / 40 km $\times$ 40 km[a] | 80 km $\times$ 40 km | 80 km $\times$ 40 km |
| Swath Width | 1920 km / 960 km[a] | 1920 km | 1920 km |
| Equator Crossing Time | 9:30 (local time) | 9:30 (local time) | 9:30 (local time) |
| Global Coverage | 1.5 days | 1.5 days | 1.5 days |
| Level 2 Processor | GDP 4.8 | GDP 4.8 | GDP 4.9 |

[a] since tandem operation with GOME-2B on $15^{th}$ July 2013.

## 2.2 Operational GOME-2 level 2 data

### 2.2.1 Total column ozone

Ozone ($O_3$) is an important trace gas in the Earth's atmosphere. In the stratosphere, ozone absorbs ultraviolet radiation from the sun, thus protecting the biosphere from harmful radiation (Eleftheratos et al., 2013; Hegglin et al., 2015). In the lower
troposphere, natural ozone has an equally important beneficial role, because it initiates the chemical removal of air pollutants and greenhouse gases from the atmosphere such as carbon monoxide (CO), nitrogen oxides ($NO_x$), and methane ($CH_4$). However, ozone at high concentration can also be harmful to humans, plants, and animals. In addition, ozone is a greenhouse gas contributes to the warming of the Earth's atmosphere. In both the stratosphere and the troposphere, ozone absorbs infrared radiation emitted from Earth's surface, trapping heat in the atmosphere. As a result, increases or decreases in stratospheric or
tropospheric ozone induce a climate forcing (Hegglin et al., 2015).

The retrieval of total column ozone from GOME-2 (ir)radiance spectra is based on the typical two step approach for weak absorbing trace gas. The first step is to apply the differential optical absorption spectroscopy (DOAS) technique (Platt and Stutz, 2008) to the GOME-2 (ir)radiance spectra within the wavelength range of 325 - 335 nm to retrieve ozone slant column densities (SCDs). Ozone absorption features are prominent in this wavelength range.In addition, GOME-2 measurements
have high signal-to-noise and manageable interference effects from other trace gases in this wavelength band. The DOAS fits includes two ozone cross-sections at 218 K and 243 K. $NO_2$ cross-section and the Ring spectrum are also included in the spectral fit.

The second step is the conversion of retrieved ozone slant column densities to vertical column densities (VCDs) using air mass factor (AMF) (Solomon et al., 1987; Palmer et al., 2001). Vertical distribution profiles are essential a priori information
to the calculation of AMF. The air mass factor calculation for ozone vertical column retrieval follows an iterative approach. The algorithm uses a standard ozone profile to retrieve an initial ozone vertical column. Based on initial result of ozone vertical column retrieval, the algorithm selects the most appropriate a priori profile from the climatology database and uses it as a priori in the next iteration. The iterations end when the change in the retrieved vertical column is less than 0.1 % or it reaches the





maximum limit of iterations. For GOME-2 total column ozone retrieval, the number of iterations is in most cases smaller than 4. The radiative transfer model, LIDORT (Spurr, 2008), is used for the radiative transfer calculation of AMF with respective to the a priori ozone profile and cloud information. Cloud parameters are retrieved from GOME-2 measurements using the OCRA and ROCINN algorithms (see section 2.2.7) and the ozone absorption inside and below the cloud is treated by the intra cloud

correction term, which is a function of solar zenith angle and cloud albedo. AMFs are computed at a single wavelength of 325.5 nm (Van Roozendael et al., 2006). As the major part of ozone is in the stratosphere which is well above clouds, therefore, all measurements (cloudy and clear sky) are used in the level 3 product. More details of the GOME-2 total column ozone retrieval can be found in Loyola et al. (2011); Hao et al. (2014).

### 2.2.2   Total and tropospheric column nitrogen dioxide

Nitrogen dioxide ($NO_2$) plays an important role in atmospheric chemistry and air quality in the troposphere. It is an air pollutant affecting human health and ecosystems. Furthermore, $NO_2$ in the troposphere is a major ozone precursor, while being a catalyst of stratospheric ozone depletion processes which is very important for climate change studies, due to its indirect effect on the global climate (Shindell et al., 2009).

The retrieval of total and tropospheric column $NO_2$ from GOME-2 follows the typical two step approach for a weak ab-

sorbing trace gas. The DOAS approach is applied to GOME-2 (ir)radiance spectra to determine $NO_2$ slant column densities at wavelength range of 425 - 450 nm (Valks et al., 2011) for GOME-2/Metop-A and GOME-2/Metop-B (GDP 4.8). $NO_2$ absorption structures are prominent and the interference effects are manageable within this spectral window. In addition, GOME-2 measurements have high signal-to-noise in this wavelength band. For GOME-2/Metop-C (GDP 4.9), an alternative fitting-window 430.2 - 465 nm is used, as there are systematic structures in the DOAS fitting residual for GOME-2C for wavelengths

<430 nm, which result in a large positive bias of ∼30 %. A single $NO_2$ reference cross-section spectrum at 240 K (Vandaele et al., 2002), and the interfering species ozone, $O_4$ and $H_2O$ as well as Ring spectrum are included in the DOAS spectral retrieval. The temperature dependence of $NO_2$ absorption cross-section has been taken into account in the air mass factor calculation to improve the accuracy of the retrieved columns.

The initial total VCD is retrieved assuming a unpolluted troposphere. Therefore, the air mass factor is weighted toward to

stratospheric $NO_2$, whereas tropospheric $NO_2$ amount is assumed to be negligible. This assumption is valid over large parts of the Earth, but in areas with significant tropospheric $NO_2$, the total column densities are underestimated and need to be corrected. The air mass factors are calculated at the mid-point of the spectral fitting window at 437.5 nm using the radiative transfer model LIDORT. A harmonic climatology of stratospheric $NO_2$ profiles is used in the air mass factor calculation to incorporate the seasonal and latitudinal variation of stratospheric $NO_2$. Tropospheric $NO_2$ columns are then derived from the

initial total columns by estimating the stratospheric content and subtracting it from the total amount. Several methods have been applied for the stratosphere $NO_2$ estimation, e.g., Boersma et al. (2007); Beirle et al. (2010). In the GDP 4.8 and 4.9, a spatial filtering approach (Wenig et al., 2004) is used by masking potentially polluted areas and then applying a low-pass filter in the zonal direction. This method shows significant improvement over the Pacific reference sector method, which assumes longitudinally homogeneous stratospheric $NO_2$ layer.





After the stratosphere-troposphere separation, the tropospheric columns can be determined using tropospheric air mass factors. Monthly average $NO_2$ profiles during GOME-2 overpass time from the chemistry transport model MOZART-2 are used for tropospheric air mass factor calculations. Cloud properties derived from GOME-2 using the OCRA and ROCINN algorithms (see section 2.2.7) are used in the calculation of air mass factors in case of (partly) cloudy conditions. The calculation

of AMF for (partly) cloudy conditions uses the independent pixel approximation. For measurements over cloudy scenes, the cloud-top is usually well above the $NO_2$ pollution in the boundary layer. When the clouds are optically thick, the enhanced tropospheric $NO_2$ concentrations cannot be detected by the satellite which can result in large errors. Therefore, GOME-2 measurements with cloud radiance fraction $>50\%$ are flagged in the level 2 data and filtered in the computation of level 3 tropospheric $NO_2$ product. More details of the GOME-2 total and tropospheric column $NO_2$ retrieval can be found in Valks

et al. (2011).

### 2.2.3 Total column water vapour

Water vapour is one of the major components in the atmosphere with strong impacts on climate and weather. Due to its abundance in the atmosphere, making it the most important natural greenhouse gas, accounting for more than $60\%$ of the greenhouse effect (Clough and Iacono, 1995; Kiehl and Trenberth, 1997). The knowledge of the spatio-temporal distribution

of water vapour over the globe is essential for weather prediction and climate assessments. Improving the understanding of variability and changes in water vapour is vital, especially considering that, in contrast to most other greenhouse gases, the distribution of water vapour is highly variable due to its short atmospheric lifetime.

The GOME-2 operational total column water vapour (TCWV) algorithm is based on the DOAS and AMF approach. The DOAS retrieval of water vapour slant columns is performed in the wavelength interval of 614 - 683 nm. Compared to other water

vapour retrieval methods, this approach focuses only on the differential absorptions, and therefore, less sensitive to instrument changes or instrument degradation issues. The algorithm consists of three basic steps (1) DOAS fit for slant column retrieval, (2) non-linearity absorption correction of slant columns and (3) slant to vertical column conversion using AMF (Wagner et al., 2003, 2006).

The DOAS fit for water vapour retrieval takes into account of $O_2$ and $O_4$ cross-sections, in addition to that of water vapour.

Three types of vegetation spectra (Wagner et al., 2007), a synthetic Ring spectrum, and inverse solar spectrum are included in the DOAS fit to improve the broadband filtering and to correct for possible offsets, e.g. caused by instrumental stray light. The retrieved water vapour slant columns are then corrected for the non-linearities arising from the fact that the fine structure water vapour absorption lines are not spectrally resolved by the GOME-2 instruments. The corrected water vapour slant columns are divided by the "measured" AMFs to convert to vertical column. The "measured" AMF is defined as the ratio between

the measured $O_2$ slant column retrieved at the same wavelength band and the known $O_2$ vertical column from the standard atmosphere. This simple approach has the advantage that it corrects in first order for the effect of albedo variation, aerosol load and cloud cover without the use of additional independent information. It is also important to note that, the GOME-2 water vapour product does not rely on additional information, except for the use of an albedo database for the AMF correction. The surface albedo used for the correction is taken from monthly albedo database derived from GOME-1 observations (Koelemeijer





et al., 2003) for high latitude (>50°), and SCIAMACHY observations (Grzegorski, 2009) at mid and low latitudes (<40°). For the transition between 40° and 50°, both products are interpolated linearly. This serves the aim to derive a climatologically relevant time series of TCWV measurements (Wagner et al., 2006; Lang et al., 2007; Noël et al., 2008). The current version of GOME-2 TCWV retrieval does not account for the terrain effect with elevated surface in the AMF correction, i.e. over

high mountain areas (>1000 m), and the retrieval errors in TCWV are significantly higher. Therefore, these measurements are flagged and not being used in the level 3 data processing.

GOME-2 measurements significantly contaminated by clouds are also flagged in the level 2 and filtered in the level 3 products. The cloud flag in the water vapour product is set based on two indicators. The first cloud flag is set if the retrieved $O_2$ slant column is below 80 % of the maximum $O_2$ slant column for the respective solar zenith angle (roughly when about

20 % from the column to ground is missing). Especially for low and medium high clouds, the relative fraction of the column shielded by clouds for $O_2$ and water vapour can be very different. The second cloud flag is set if cloud fraction and cloud top albedo exceeds 0.6. More details of the GOME-2 total column water vapour retrieval can be found in Grossi et al. (2015).

### 2.2.4   Total column bromine monoxide

Bromine monoxide (BrO) in the lower stratosphere is involved in chain reactions that deplete ozone (Wennberg et al., 1994),

while in the troposphere BrO changes the oxidizing capacity through the destruction of ozone. In particular, large amounts of BrO are often observed in the polar boundary layer during spring-time (Platt and Wagner, 1998; Richter et al., 1998), known as "bromine explosion", and lead to severe tropospheric ozone depletion by autocatalytic reactions. In addition to polar sea ice regions, enhanced BrO concentrations were also detected over salt lakes/marshes (Hebestreit et al., 1999; Hörmann et al., 2016), in the marine boundary layer, and in volcanic plumes (Theys et al., 2009; Hörmann et al., 2013).

The GOME-2 total BrO retrieval is also a typical DOAS and AMF algorithm. The DOAS retrieval of BrO slant columns is applied to the spectral range of 332 - 359 nm which covers five BrO absorption peaks and minimized the interference from other trace gases, especially formaldehyde (Theys et al., 2011). This fitting window can also minimize other artefacts due to instrument noise, viewing angle dependency and interference from incomplete ring effect correction.

The instrumental degradation of GOME-2 has negative influences on the spectral fit and results in higher residuals. Thus,

affecting the noise level in the BrO columns, and the average slant columns values. Therefore, an equatorial offset correction is applied on a daily basis to the BrO data (Richter et al., 2002). This correction reduces the influences of the instrumental degradation on the total BrO column time series. Averaged BrO slant columns in the tropical latitudinal band between ±5° are calculated on a daily basis, assuming small equatorial BrO columns with no significant seasonal variations. These averaged slant columns are then subtracted from all slant columns and a constant equatorial slant column offset of $7.5 \times 10^{13}$ molec/cm$^2$ is

added. Corrected BrO slant columns are then converted to vertical columns by using AMFs. In the GOME-2 total column BrO retrieval, AMFs are calculated at 344 nm (mid-point of DOAS fitting range) using the radiative transfer model LIDORT (Spurr, 2008). Monthly climatology BrO profiles from the chemistry transport model SLIMCAT (Chipperfield, 1999; Bruns et al., 2003) are used in the AMF calculations. In case of (partly) cloudy cases, GOME-2 cloud properties determined with the OCRA and ROCINN algorithms (see section 2.2.7) are used to calculate the air mass factors in association with the independent pixel





approximation. As BrO has major contribution from the stratosphere which is usually above clouds, therefore, all measurements (cloudy and clear sky) are used in the level 3 product. More details of the GOME-2 total column BrO retrieval can be found in Theys et al. (2011).

### 2.2.5 Total column formaldehyde

5 Formaldehyde (HCHO) is an intermediate product of the oxidation of almost all volatile organic compounds (VOCs). Therefore, it is widely used as an indicator of non-methane volatile organic compounds (NMVOCs) (Fried et al., 2011). VOCs also have significant impacts on the abundance of hydroxyl (OH) radicals in the atmosphere, which is the major oxidant in the troposphere. Major HCHO sources over continents include the oxidation of VOCs emitted from plants, biomass burning, traffic and industrial emissions. Oxidation of methane ($CH_4$) emitted from the ocean is the main source of HCHO over water and 10 pristine continental areas.

The GOME-2 total HCHO column retrieval algorithm also follows the two steps approach with DOAS retrieval of HCHO slant columns and subsequently converts the slant columns to vertical columns by using AMFs. To reduce the interference between HCHO and BrO absorption features, a two-step DOAS retrieval of HCHO slant columns is used (De Smedt et al., 2012). The first step is to determine BrO slant columns with a larger fitting window of 332 - 359 nm which includes five BrO 15 absorption peaks and effectively minimized the cross-correlation between BrO and HCHO. The retrieved BrO slant columns are then fixed in the subsequent DOAS retrieval of HCHO slant columns in the spectral band of 328.5 - 346 nm.

Although the DOAS fit settings are optimized to minimize interference from other factors, there are still unresolved spectral absorption interferences between HCHO and BrO and results in obvious zonally and seasonally dependency. In order to reduce the impact of the artefacts, an absolute normalisation is applied to HCHO slant columns on a daily basis using the reference 20 sector method (Khokhar et al., 2005). The reference sector is chosen over the Pacific Ocean (Longitude: 140° - 160°W), where the only source of HCHO is the oxidation of $CH_4$ which can be reproduced by model simulation quite well. The mean HCHO slant column density in the reference sector is determined by a polynomial fit, which is then subtracted from the retrieved slant columns on this day, and replaced by a HCHO background value taken from IMAGES version 2 chemistry transport model simulations (Müller and Stavrakou, 2005).

25 Corrected HCHO slant columns are then converted to vertical columns by using AMFs. HCHO AMFs are calculated at 335 nm using the radiative transfer model LIDORT. Monthly averaged profiles taken from chemistry transport model (CTM) IMAGES version 2 simulation in 2007 are used as a priori HCHO profiles in the AMF calculations. The IMAGES version 2 model simulations are in a horizontal resolution of 2.0° (latitude) × 2.5° (longitude), with 40 vertical layers extending from the surface up to ∼44 hPa. In case of the presence of clouds, cloud properties derived by the OCRA and ROCINN 30 algorithms (see section 2.2.7) are used to calculate the air mass factors using the independent pixel approximation. For cloudy scene measurements, clouds are usually above the boundary layer where the major part of HCHO is located. If the clouds are optically thick, HCHO below cloud cannot be detected by the satellite and results large uncertainties. Therefore, measurements with cloud radiance fraction >50 % are flagged and not being used in the level 3 data processing. More details of the GOME-2 total column HCHO retrieval can be found in De Smedt et al. (2012).





### 2.2.6  Total column sulphur dioxide

Sulphur dioxide ($SO_2$) is an important trace species playing key role in atmospheric chemistry on both local and global scales through the formation of sulphate aerosols and sulphuric acid. The impacts of $SO_2$ range from short-term pollution to climate forcing. $SO_2$ emits to the atmosphere through both natural and anthropogenic processes. About one-third of the global sulphur emissions originate from natural sources (volcanoes and biogenic dimethyl sulphide), the major contributors to the total budget are anthropogenic emissions through the combustion of fossil fuels (coal and oil) and smelting.

The GOME-2 $SO_2$ retrieval algorithm also follows the two steps approach with DOAS retrieval of $SO_2$ slant columns and subsequently converts the slant columns to vertical columns by using AMFs. The DOAS algorithm for $SO_2$ is based on the DOAS fit settings dedicated for ozone retrieval with adjustments to optimize for $SO_2$ retrieval (Rix et al., 2009, 2012). The DOAS fit for the retrieval of $SO_2$ slant column is applied to the wavelength range of 315 - 326 nm for GOME-2/Metop-A and -B (GDP 4.8) and 312 - 325nm for GOME-2/Metop-C (GDP 4.9).

The background level of atmospheric $SO_2$ is very low over large parts of the Earth. To account for any systematic bias in the retrieved total column $SO_2$ and to ensure a geospatial consistency of the results, a background correction is applied to the data to avoid non-zero columns over regions known to have very low $SO_2$ and at high solar zenith angles. The background correction scheme calculates offset $SO_2$ slant columns based on latitude and surface height information. This offset is calculated on a daily basis with measurements binned to $2°$ resolution in latitude. In addition, the offset values are calculated separately at 5 surface altitude bins. Median offset values based on the calculated offset values in the last two weeks before the day of interest is used as the corresponding offset $SO_2$ slant columns to minimize the influences from outliers or missing data in the daily dataset. This latitude and altitude dependent value is then subtracted from the $SO_2$ slant column derived from the DOAS retrieval.

Corrected $SO_2$ slant columns are then converted to vertical columns by using AMFs. The major challenge in the $SO_2$ retrieval is that the vertical distribution of $SO_2$ in the atmosphere is usually unknown. Depending on the type of emission, $So_2$ can be located from the ground up to the stratosphere. In the GOME-2 $SO_2$ retrieval, it assumes that most of the atmospheric $SO_2$ is emitted from volcanic related activities. In the AMF calculation, the $SO_2$ plume is assumed to follow a Gaussian profile shape with central plume height of 6 km a.s.l. (at about 500 hPa). The AMF is calculated at 320 nm (GOME-2A and GOME-2B) and 313 nm (GOME-2C) using the radiative transfer model LIDORT. For scenarios in the presence of clouds, GOME-2 cloud properties determined with the OCRA and ROCINN algorithms are used to calculate the air mass factors.

### 2.2.7  Cloud parameters

It is very important to derive cloud properties from GOME-2 observations as clouds significantly affects the retrieval of tropospheric trace gases. The most predominant effect of clouds in trace gas retrieval is that they shield the trace gas absorption below clouds. However, clouds can also enhance the absorption due to multiple scattering within the cloud. The GOME-2 retrieval of the trace gases vertical columns assumes independent pixel approximation (IPA) for cloudy scene measurements. For tropospheric species, i.e., tropospheric $NO_2$, water vapour, HCHO, especially in the case of low clouds and large cloud





fractions, the presence of clouds can result in large errors. Therefore, measurements with high cloud cover are flagged in these products and being filtered in the level 3 process.

The optical cloud recognition algorithm (OCRA) and retrieval of cloud information using neural network (ROCINN) algorithms (Loyola et al., 2007; Lutz et al., 2016) are used to obtain cloud information from GOME-2 observations. Clouds are

treated as reflecting Lambertian surfaces in the algorithm and cloud information is reduced to the specification of three parameters: cloud fraction, cloud-top albedo and cloud-top pressure. The radiometric cloud fraction is retrieved from the broad-band polarization of UVN measurements (UV-VIS-NIR) by OCRA, while effective cloud pressure and cloud albedo are retrieved by ROCINN from observations in the oxygen A-band ($O_2$-A) around 760 nm. The OCRA algorithm separates each measurement to two components: a cloud-free background and a residual contribution describing the influence of clouds. The key to the algo-

rithm is the construction of a cloud-free composite that is invariant with respect to atmosphere, to topography and to solar and viewing geometries. The effective cloud fraction is determined by examining the separation between the reflectance measured by the PMDs of GOME-2 and their cloud-free composite values. Note that OCRA is also sensitive to scattering by aerosols present in a given GOME-2 scene. Therefore, the retrieved cloud fraction also includes the aerosol effects. Furthermore, the GOME-2 cloud algorithm has been improved to distinguish clouds for measurements affected by sun-glint over ocean, which

is a common phenomenon that occur at the edges of the GOME-2 swath. The detection of cloud over sun-glint is achieved by analyzing the broad-band polarization measurements (Loyola et al., 2011; Lutz et al., 2016).

The cloud fraction derived from OCRA is then ingested by ROCINN as fixed input (Loyola et al., 2007), which derived cloud-top height and cloud albedo using measurements at the $O_2$-A band. In the radiative transfer simulations, oxygen absorption in the earthshine spectra including the reflection from Earth's surface and cloud-top is considered in the atmospheric

radiative transfer. Surfaces are assumed to be Lambertian reflectors. Black-sky albedo climatology from the MEdium Resolution Imaging Spectrometer (MERIS) is used as input for the radiative transfer and in ROCINN version 3. Radiative transfer simulations in ROCINN include Rayleigh scattering and polarization. High-resolution reflectances computed with VLIDORT (Spurr, 2006) are used to create a complete data set of simulated reflectance for all viewing geometries and geophysical scenarios, and for various combinations of cloud fraction, cloud-top height and cloud-top albedo. The inversion is performed using

neural network techniques. The cloud-top height retrieved by ROCINN is converted to cloud-top pressure assuming a U.S. standard atmosphere (Anderson et al., 1986). The retrieved cloud properties are then used in the subsequent processing of trace gas column retrieval and provided in the corresponding level 3 products.

## 2.3 Validation data sets

### 2.3.1 Brewer ozone measurements

Brewer ozone data are obtained the World Ozone and Ultraviolet Radiation Data Center (WOUDC, http://www.woudc.org). The WOUDC data center is part of the Global Atmosphere Watch (GAW) programme of the World Meteorological Organization (WMO), providing quality-assured Brewer measurements. Brewer instruments measure intensity at several wavelength intervals in the UV band. Total column ozone is retrieved from the relative intensities among these UV channel. Brewer ozone



data has long been used to validate satellite observations of ozone (Balis et al., 2007a, b; Antón et al., 2009; Loyola et al., 2011; Koukouli et al., 2012, 2015; Garane et al., 2018, 2019). In this study, we only use the direct sun Brewer observations of total column $O_3$ for the validation of GOME-2 level 3 product.

### 2.3.2 ZSL-DOAS and MAX-DOAS $NO_2$ measurements

Zenith-Scattered-Light Differential Optical Absorption Spectroscopy (ZSL-DOAS) data are obtained from the Network for the Detection of Atmospheric Composition Change (NDACC). NDACC ZSL-DOAS network provides total column $NO_2$ observations with standardized operating procedures and harmonized retrieval methods. ZSL-DOAS data from NDACC stations is available on the NDACC data host facility (see http://www.ndacc.org). ZSL-DOAS measurements during twilight periods are sensitive to stratospheric absorbers due to the geometrical enhancement of the optical path in the stratosphere. Therefore, it has
long been used for the validation of satellite total $NO_2$ observations (Ionov et al., 2008; Celarier et al., 2008). The retrieval of total column $NO_2$ from ZSL-DOAS observations is based on the Langley method, which calculates the corresponding air mass factor according to its observation and solar geometry. As most of the ZSL-DOAS sites are located in relative clean regions, therefore, the major contribution of total column $NO_2$ is expected to be coming from the stratosphere. Due to the morning overpass time of GOME-2, ZSL-DOAS observations of total column $NO_2$ during the morning twilight period are used to val-
idate GOME-2 level 3 total $NO_2$ products. As the measurement time of GOME-2 and ZSL-DOAS are nigh, therefore, these data are comparable without the need of photochemical correction.

The Multi-AXis Differential Optical Absorption Spectroscopy (MAX-DOAS) is a passive remote sensing technique which uses spectroscopic observations of scattered sunlight at different viewing zenith angles to derive column densities of trace gas. Due to its compact experimental setup and high sensitivity to lower troposphere, it has been widely used for the validation of
satellite observations of tropospheric column $NO_2$ (Brinksma et al., 2008; Celarier et al., 2008; Irie et al., 2008, 2009, 2012, 2016; Ma et al., 2013; Kanaya et al., 2014; Chan et al., 2015, 2018, 2019, 2020b; Drosoglou et al., 2017; Wang et al., 2017; Compernolle et al., 2020; Pinardi et al., 2020a; Verhoelst et al., 2021). Ground-based MAX-DOAS instruments are operated by various research institutes around the world, and the data is centrally managed by BIRA-IASB within the context of NItrogen Dioxide and FORmaldehyde VALidation (NIDFORVAL). The affiliation of MAX-DOAS instruments in the NDACC network
is still under progress, following efforts done in the NORS, QA4ECV and ESA's FRM4DOAS project to harmonize and automatize data processing. In this study, MAX-DOAS observations of tropospheric column $NO_2$ are used to validate GOME-2 level 3 tropospheric $NO_2$ products.

### 2.3.3 Sun-photometer water vapour measurements

The AERosol RObotic NETwork (AERONET) uses CIMEL CE-318 sun-photometers to measure direct sun and sky radiance
at multiple wavelengths (Holben et al., 1998). These sun-photometer observations do not only provide information of aerosol optical properties (Holben et al., 2001) but also of columnar water vapour content (Alexandrov et al., 2009). Water vapour columns are retrieved from sun-photometer observations in the near infrared (NIR) at 940 nm where water vapour absorption is rather strong. The inversion of water vapour columns is based on the attenuation of radiation through the atmosphere. A





more detailed description of the water vapour retrieval algorithm can be found in Alexandrov et al. (2009). In total, there are over 1000 AERONET stations around the globe providing columnar water vapour observations and they have been used extensively for satellite validation (Bennouna et al., 2013; Diedrich et al., 2015; Martins et al., 2019; Chan et al., 2020a; Garane et al., 2022). The AERONET water vapour product has also been validated by microwave radiometry, GPS and radiosondes

measurements (Pérez-Ramírez et al., 2014). The sun-photometer measurements are in general underestimating the columnar water vapour by 6 - 9 % (Pérez-Ramírez et al., 2014). In this study, cloud screened and quality assured level 2.0 data are used to validate GOME-2 level 3 total column water vapour products.

### 2.3.4   ZSL-DOAS BrO measurements

The ZSL-DOAS observations at Harestua (60.22 °N, 10.75 °E), Norway are used to validate the GOME-2 level 3 total column

BrO product. ZSL-DOAS observation of total BrO columns are photochemically corrected to the GOME-2 overpass time (9:30 local time). The operation of the ZSL-DOAS instrument and the retrieval of BrO column are performed by BIRA-IASB. Detailed description of the ZSL-DOAS instrument setup and BrO column retrieval algorithm can be found in Hendrick et al. (2007).

### 2.3.5   MAX-DOAS HCHO measurements

Ground-based MAX-DOAS observations are used to validate GOME-2 level 3 total column HCHO product. MAX-DOAS observations show very good sensitivity in the troposphere where most of the HCHO resides. Therefore, it has long been used for satellite validation (Vigouroux et al., 2009; Li et al., 2013; De Smedt et al., 2015b, 2021; Wang et al., 2017; Chan et al., 2019, 2020b; Kumar et al., 2020). The retrieval of HCHO columns from MAX-DOAS observations are performed within a wavelength range similar to the GOME-2 retrieval, i.e., 328 - 359 nm. Ground-based MAX-DOAS instruments are

operated by various research institutes around the world, and the data is centrally managed by BIRA-IASB within the context of NIDFORVAL.

### 2.3.6   Pandora $SO_2$ measurements

The Pandonia Global Network is a direct-sun spectrometer network used to monitor trace gas worldwide. The Pandora instrument is used to measure columnar amounts of trace gases in the atmosphere. Pandora determines trace gas amounts from

direct-sun observations by using the DOAS technique with theoretical solar spectrum as a reference. As the anthropogenic $SO_2$ emission has been reduced significantly in the recent decades, the background $SO_2$ level is mostly zero around the globe and only few locations with signification anthropogenic $SO_2$ sources. Considering the low background $SO_2$ level and the high measurement noise of $SO_2$ data, it is more appropriate to validate the satellite observations over locations with significant variation and sources. Mexico City is one of the few places with significant anthropogenic $SO_2$ sources. Therefore, we use the

Pandora $SO_2$ observations at Mexico City to validate GOME-2 level 3 total column $SO_2$ products.



## 3 Methodology

### 3.1 Gridding algorithm

GOME-2 level 3 data products are developed with the aim of providing easily translatable data sets to both facilitate scientific progress (e.g. on climate trend analysis and low-frequency climate variability) and satisfy public interest. The processing of
5 GOME-2 level 3 data requires binning of the level 2 data onto a regular 2-dimensional latitude-longitude grid.

The binning of level 2 data to regular latitude-longitude grid includes taking the arithmetic mean and standard deviation of all level 2 data points falling onto the grid cell in a given period, i.e., a day or a month, with possible trimming of low quality measurements due to large spectral fit residual and cloud contamination for tropospheric species, i.e., tropospheric $NO_2$, water vapour and HCHO. For all species, only forward-scan pixels are used in the gridding process. In case of cloudy measurements,
most of the tropospheric gases, i.e., tropospheric $NO_2$, water vapour and HCHO, are mainly situated below clouds, while satellite observations could not measure the part below cloud and result in large uncertainties. Therefore, these measurements are not used in the production of level 3 data. For stratospheric species, i.e., total column $O_3$, $NO_2$, BrO and $SO_2$, no cloud filtering is applied.

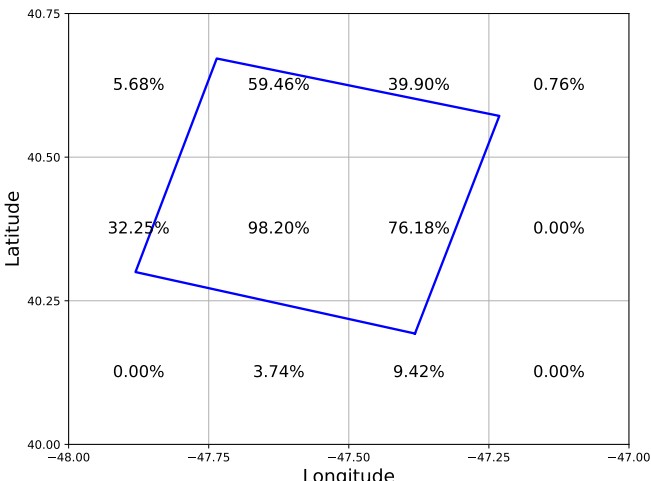

**Figure 1.** A GOME-2A ground pixel (blue) overlaid on a $0.25° \times 0.25°$ latitude-longitude grid (gray). The percentage of overlap (weighting) for each grid box is indicated.

Several gridding routines have been developed to create global and regional maps of trace gas distribution, e.g., Wenig
et al. (2008); Chan et al. (2012); Kuhlmann et al. (2014). These gridding algorithms typically assume that measurement values are constant within the satellite pixel boundaries. This assumption is considered sufficient for creating global maps. More sophisticated approach uses parabolic spline method to interpolated adjacent satellite pixels to create high resolution (e.g., $1\,km \times 1\,km$) regional maps (Kuhlmann et al., 2014). As GOME-2 ground pixel size is relatively large, a significant grid





effect would be induced by assigning each GOME-2 measurement to a single grid cell based on their center coordinates of the GOME-2 ground pixel, without taking into account the pixel geometry and extension. Therefore, the gridding process considers the overlapping area of the GOME-2 ground pixel and the latitude-longitude grid. For grid cells partially overlapped with the satellite pixel, the percentage of overlap (satellite pixel fully covers the entire grid cell is considered as 100 % overlap)

is calculated and used as weighting for the calculation of mean value, uncertainty and standard deviation. Figure 1 shows an example of the calculation of the weighting (percentage of overlap) for grid boxes overlapping with a GOME-2A ground pixel. The gridded columns can be expressed as Eq. 1.

$$VCD_g = \frac{\sum_{i=1}^{n} VCD_i \times w_i}{\sum_{i=1}^{n} w_i} \tag{1}$$

where $VCD_g$ is the gridded trace gas column while $VCD_i$ represents each individual satellite measurement (partly) overlap-

ping with the grid cell. The weighting is denoted as $w$ which is the percentage of the grid cell covered by the satellite pixel. The uncertainty of gridded columns can be express as Eq. 2.

$$E_g = \sqrt{\frac{\sum_{i=1}^{n} E_i^2 \times w_i^2}{\sum_{i=1}^{n} w_i^2}} \tag{2}$$

where $E_g$ is the uncertainty of gridded trace gas column while $E_i$ represents the uncertainty of each individual measurement. The standard deviation of gridded columns can be express as Eq. 3.

$$\sigma_g = \sqrt{\frac{\sum_{i=1}^{n} VCD_i^2 \times w_i^2}{\sum_{i=1}^{n} w_i^2} - \left(\frac{\sum_{i=1}^{n} VCD_i \times w_i}{\sum_{i=1}^{n} w_i}\right)^2} \tag{3}$$

where $\sigma_g$ is the standard deviation of gridded trace gas column.

## 3.2   Sampling resolution

The processing of GOME-2 level 3 data requires binning of the level 2 data onto a regular 2-dimensional latitude-longitude grid. The selection of appropriate resolution of the latitude-longitude grid is essential for the production of level 3 products.

On one hand, it is important to preserve the original spatial features captured in the level 2 data with higher spatial resolution, but on the other hand, it is necessary to keep the data files in a reasonable size to be user friendly.

To select the best spatial resolution for the level 3 product, we have analyzed the binning results with various resolutions, i.e., $0.1° \times 0.1°$, $0.25° \times 0.25°$ and $0.5° \times 0.5°$. Figure 2 shows GOME-2A data of each trace gas species gridded in different resolutions and level 2 data in the original instrument resolution for an orbit over North China on $15^{th}$ July 2014. Missing

data is mainly due to filtering of cloudy pixels and other low quality observations. GOME-2A data is shown due to its highest spatial resolution among all three GOME-2 instruments (GOME-2A: $40\,km \times 40\,km$ after $15^{th}$ July 2013, GOME-2B and C: $40\,km \times 80\,km$). We looked into the spatial smoothing/averaging effect over North China, as this region is expected to show



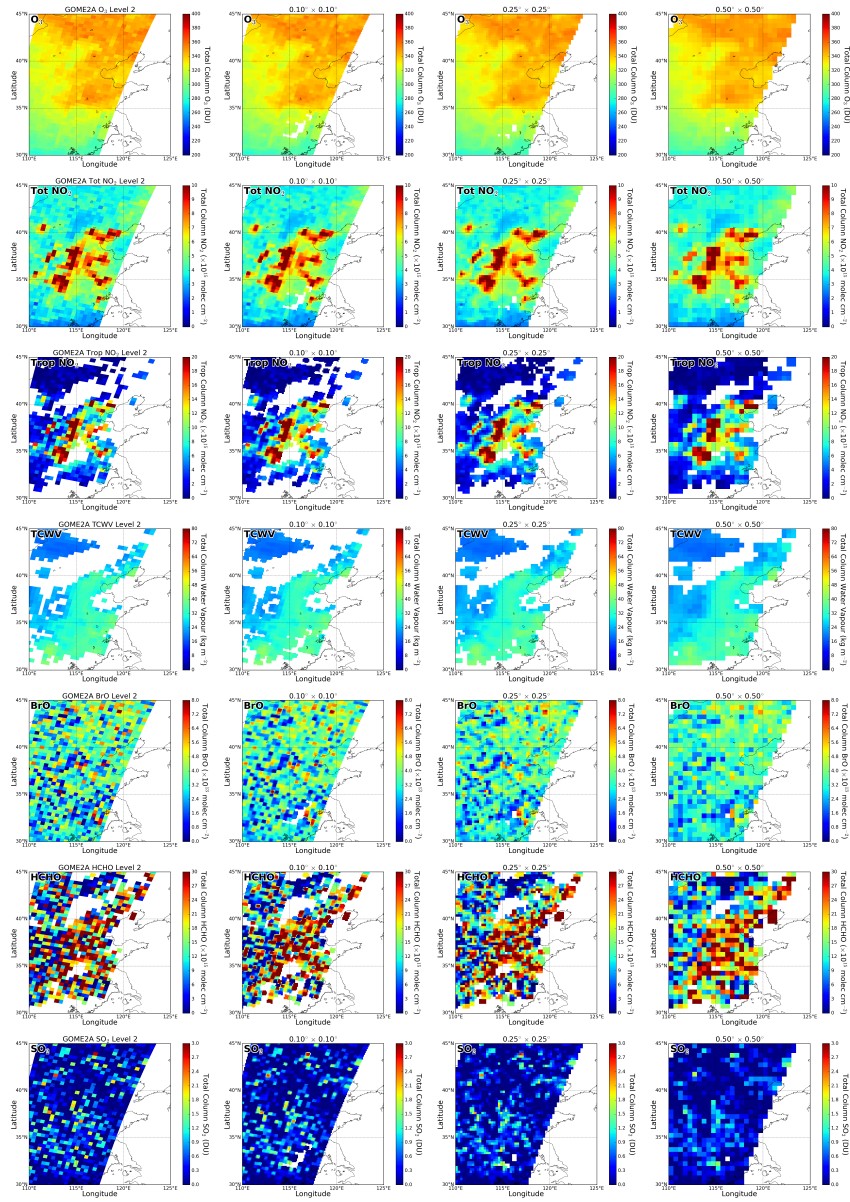

**Figure 2.** GOME-2A observations of total column $O_3$ ($1^{st}$ row), total column $NO_2$ ($2^{nd}$ row), tropospheric column $NO_2$ ($3^{th}$ row), total column water vapour ($4^{th}$ row), total column BrO ($5^{th}$ row), total column HCHO ($6^{th}$ row), and total column $SO_2$ ($7^{th}$ row). Data are shown in the original instrument resolution ($1^{st}$ column from the left), gridded with $0.1° \times 0.1°$ resolution ($2^{nd}$ column from the left), $0.25° \times 0.25°$ resolution ($3^{th}$ column from the left), and $0.5° \times 0.5°$ resolution (column on the right). GOME-2A observations on 15 July 2014 over North China are shown. Missing data are mainly due to cloudiness.



strong spatial gradients of tropospheric pollutants, i.e., $NO_2$. Data in all four resolutions show very similar spatial structures. The absolute values of level 3 data are also consistent with the level 2 product. The results show gridding GOME-2 data with higher spatial resolution (i.e., $0.1° \times 0.1°$) better preserve the original GOME-2 instrument footprint, while rather strong smoothing/averaging effect is observed from data gridded with lower spatial resolution (i.e., $0.5° \times 0.5°$). Although gridded

data with $0.25° \times 0.25°$ resolution shows some smoothing/averaging effect, it still captures the spatial variations reasonably well.

Figure 3 shows monthly averaged GOME-2A data of each trace gas species gridded in different resolutions over North China in July 2014. Differences between data gridded with different resolutions are also shown for reference. Data gridded in all three resolutions show very similar spatial structures. Hotspots of anthropogenic pollutants, i.e., tropospheric $NO_2$, can

be clearly observed from the monthly averaged data. Species with major contribution from natural sources, e.g., $O_3$ and water vapour, show rather smooth appearance. Despite large numbers of observations are included in the monthly averaging process, species with lower signal to noise ratio, e.g., HCHO and $SO_2$, still show rather high background noise. This is mainly due to the low column density and absorption of these species. This effect is as expected more significant for data gridded in higher spatial resolution, i.e., $0.1° \times 0.1°$, due to less spatial averaging. Traces of the satellite footprints can still be seen in the

$0.1° \times 0.1°$ resolution monthly averaged data, while the satellite footprints are much less significant in the $0.25° \times 0.25°$ and $0.5° \times 0.5°$ resolution data. The differential plots between data gridded with $0.1° \times 0.1°$ and $0.25° \times 0.25°$ resolution in general show only very small differences. Slightly larger discrepancies mainly appear over pollution hotspots, i.e., for tropospheric $NO_2$. In contrast, data in $0.5° \times 0.5°$ resolution show much bigger differences from $0.1° \times 0.1°$ resolution data. Compared to $0.25° \times 0.25°$ resolution data, $0.5° \times 0.5°$ resolution data shows 2 to 4 times higher underestimation of tropospheric $NO_2$

columns over pollution hotspots. The comparison of GOME-2 data gridded in different resolutions indicates that $0.25° \times 0.25°$ resolution is a balance to preserve the satellite resolution (GOME-2A: $40\,km \times 40\,km$, GOME-2B and C: $40\,km \times 80\,km$) while capturing the strong spatial variations for most of the tropospheric gases, i.e., $NO_2$, water vapour and HCHO. In addition, the data file size of level 3 products with $0.1° \times 0.1°$ resolution is about 6 times larger than that of $0.25° \times 0.25°$, while the information content does not show significant difference, especially for monthly products. Therefore, we concluded that

$0.25° \times 0.25°$ resolution is a suitable choice for GOME-2 level 3 products.

### 3.3 Verification and Validation Methods

The GOME-2 level 3 products are generated from the level 2 datasets which have already been fully validated (see validation reports in https://acsaf.org/valreps.php). Therefore, the verification and validation of GOME-2 level 3 product mainly focus on two major aspects, the consistency among the three GOME-2 sensors and the comparison to reference ground-based mea-

surements. Each GOME-2 level 3 product is compared to different reference ground-based measurements, information of the reference ground-based measurements used to validate GOME-2 level 3 products are listed in Table 2.

The comparison of GOME-2 level 3 data to reference ground-based measurements requires spatial and temporal matching of the two data sets. The following criteria are applied to co-locate the GOME-2 level 3 products and ground-based reference data sets.

**Figure 3.** Monthly averaged GOME-2A observations of total column $O_3$ ($1^{st}$ row), total column $NO_2$ ($2^{nd}$ row), tropospheric column $NO_2$ ($3^{th}$ row), total column water ($4^{th}$ row), vapour total column BrO ($5^{th}$ row), total column HCHO ($6^{th}$ row), and total column $SO_2$ ($7^{th}$ row) over North China in July 2014. Gridded data with $0.1° \times 0.1°$ resolution ($1^{st}$ column from the left), $0.25° \times 0.25°$ resolution ($2^{nd}$ column from the left), and $0.5° \times 0.5°$ resolution ($3^{th}$ column from the left) are shown. Differences between $0.1°$, $0.25°$ and $0.5°$ are also shown for reference.



**Table 2.** Summary of the reference ground-based measurements used to validate GOME-2 level 3 products.

| GOME-2 Product | Reference Measurement | Remark |
| --- | --- | --- |
| Total Column $O_3$ | Brewer | see Section 2.3.1 |
| Total Column $NO_2$ | ZSL-DOAS | see Section 2.3.2 |
| Tropospheric Column $NO_2$ | MAX-DOAS | see Section 2.3.2 |
| Total Column Water Vapour | Sun-photometer | see Section 2.3.3 |
| Total Column BrO | ZSL-DOAS | see Section 2.3.4 |
| Total Column HCHO | MAX-DOAS | see Section 2.3.5 |
| Total Column $SO_2$ | Pandora | see Section 2.3.6 |

- The grid cell of the level 3 GOME-2 products covering the ground-based measurement site is paired with the daily/monthly ground-based measurements

- For ground-based Brewer, MAX-DOAS, sun-photometer and Pandora measurements, they are temporally averaged around the GOME-2 overpass time from 8:30 to 10:30 (local time)

- For ZSL-DOAS measurements, morning twilight period measurement is used for comparison

After co-locating the GOME-2 and ground-based datasets, we compare the GOME-2 level 3 products to reference ground-based data sets through scatter plot, histogram of the differences, and sort the differences/bias by year, latitude band or measurement site as box plot and time series to investigate the systematic bias/error.

## 4  GOME-2 level 3 products

The GOME-2 level 3 products are in two different temporal resolution, daily and monthly. Both daily and monthly level 3 product consists of gridded trace gas columns and other auxiliary parameters, i.e., cloud, surface and statistical parameters. The level 3 products are separated for each species (i.e., $O_3$, $NO_2$, water vapour, BrO, HCHO and $SO_2$) and each GOME-2 instrument (i.e., GOME-2A, B and C). All products are in a spatial resolution of $0.25° \times 0.25°$ with coordinates ranging from 180° W to 180° E in longitude and from 90° S to 90° N in latitude (720 (latitude) $\times$ 1440 (longitude) grid cell). The

data are organized in a user-friendly and self-describing NetCDF-4 (Network Common Data Form) format, based upon the instrument/platform (GOME-2A/Metop-A, GOME-2B/Metop-B or GOME-2C/Metop-C) and the temporal period of collection (daily or monthly data set).

Figure 4 shows an example of the daily level 3 product for all trace gases and all GOME-2 instruments, while example of monthly level 3 data is shown in Figure 5. Missing data are mainly due to filtering of low quality data, e.g., cloud contamination,

high solar zenith angle and high spectral fit residual. The spatial coverage of GOME-2A daily product is different from GOME-2B and C due to the improvement of spatial resolution after it went in tandem operation with GOME-2B in July 2013. The noise level of monthly GOME-2A data is significantly higher than that of GOME-2B and C and it is mainly related to less

**Figure 4.** Daily level 3 product of GOME-2A ($1^{st}$ column), GOME-2B ($2^{nd}$ column), and GOME-2C ($3^{th}$ column) for 15 January 2020. Total column $O_3$ ($1^{st}$ row), total column $NO_2$ ($2^{nd}$ row), tropospheric column $NO_2$ ($3^{th}$ row), total column water vapour ($4^{th}$ row), total column BrO ($5^{th}$ row), total column HCHO ($6^{th}$ row), and total column $SO_2$ ($7^{th}$ row) are shown.

**Figure 5.** Monthly level 3 product of GOME-2A ($1^{st}$ column), GOME-2B ($2^{nd}$ column), and GOME-2C ($3^{th}$ column) for January 2020. Total column $O_3$ ($1^{st}$ row), total column $NO_2$ ($2^{nd}$ row), tropospheric column $NO_2$ ($3^{th}$ row), total column water vapour ($4^{th}$ row), total column BrO ($5^{th}$ row), total column HCHO ($6^{th}$ row), and total column $SO_2$ ($7^{th}$ row) are shown.



spatial averaging and instrument aging. This effect is particularly obvious for species with lower signal to noise ratio, e.g., HCHO and $SO_2$. In addition, the stripe pattern is also more significant for GOME-2A, e.g., water vapour product, due to the narrower swath width of GOME-2A measurements.

## 5 Validation

In this section, we present validation results of the GOME-2 level 3 products. The GOME-2 level 3 products are first examined with respect to their cross-sensor consistency. In addition, level 3 products of each trace gas are compared to ground-based observations for validation.

### 5.1 Cross-sensors consistency

#### 5.1.1 Average and bias

Figure 6 shows the global monthly mean time series of (a) total column $O_3$, (b) total column $NO_2$, (c) tropospheric column $NO_2$, (d) total column water vapour, (e) total column BrO, (f) total column HCHO and (g) total column $SO_2$ for GOME-2A, B and C. The error bars represent the $1\sigma$ standard deviation of variation. All species except for $SO_2$ show pronounced seasonal variation patterns. The seasonal patterns are related to the natural variability and the variation of coverage area of the GOME-2 measurements. The global monthly mean total column $O_3$ time series of GOME-2A, B and C are mostly overlapping with each

other, indicating the good agreement among the three sensors. However, GOME-2C is reporting a slightly higher (2 - 3 DU) value compared to GOME-2A and B. This is likely related to the small difference in instrument characteristic, e.g., scan angle dependency and polarization sensitivity. For total column $NO_2$, observations from GOME-2A and B show very good consistency, while GOME-2C data are about $1.2 \times 10^{14}$ molec cm$^{-2}$ higher than that of GOME-2A and B. Tropospheric column $NO_2$ from GOME-2A and B are also in good agreement. However, GOME-2C observations are about $1.5 \times 10^{14}$ molec cm$^{-2}$

lower than GOME-2A and B observations. The discrepancies in $NO_2$ observations is likely related to the different processor versions (GDP 4.8 for GOME-2A and B and GDP 4.9 for GOME-2C). The spectral fitting band of $NO_2$ is slightly different in different processor version (see Section 2.2.2). Previous validation study shows that the $NO_2$ slant columns retrieved from GOME-2C observations are slightly higher than that of GOME-2B (Pinardi et al., 2019), indicating the impact of the different spectral fitting bands on the $NO_2$ retrieval. In addition, the positive bias in the GOME-2C total column $NO_2$ shows an

impact on the tropospheric columns in the stratospheric and tropospheric separation process (Pinardi et al., 2019), and results the discrepancies in the tropospheric columns. Total column water vapour measurements from all three GOME-2 sensors also show very good consistency with bias smaller than 1 kg m$^{-2}$. For BrO observations, GOME-2B measurements show a negative bias of $\sim$1.0 - 1.5 $\times 10^{12}$ molec cm$^{-2}$ compared to GOME-2A and C. The discrepancies are partly related to the difference in the scanning swath width and the scan angle dependency (Merlaud et al., 2020). The impact of scan angle dependency

on BrO measurements is more significant for GOME-2C compared to GOME-2B, which is likely linked to the polarization sensitivity of the GOME-2C instrument (Merlaud et al., 2020). GOME-2A observations of total column HCHO are in general

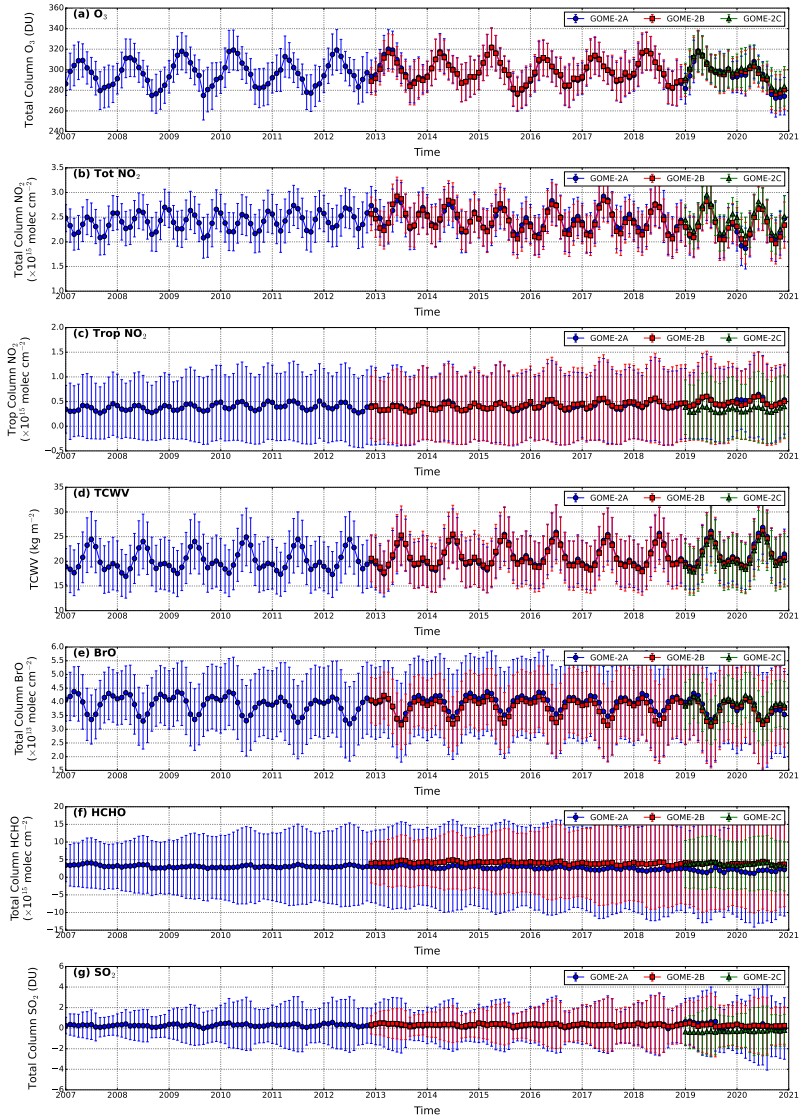

**Figure 6.** Time series of global monthly mean (a) total column $O_3$, (b) total column $NO_2$, (c) tropospheric column $NO_2$, (d) total column water vapour, (e) total column BrO, (f) total column HCHO and (g) total column $SO_2$ for GOME-2A (blue lines), GOME-2B (red lines) and GOME-2C (green lines). The error bars represent the $1\sigma$ standard deviation variation.

$1.5 - 1.9 \times 10^{12}$ molec cm$^{-2}$ lower than GOME-2B and C measurements. Lower HCHO columns are observed by GOME-2A over Amazon, Central Africa, Southeast Asia and Australia (see Figure 5), thus results slightly lower global averages. Similar to BrO measurements, the scan angle dependency issue is also reported to be significant for GOME-2C HCHO observations (Pinardi et al., 2020b). The scan angle dependency effect can also be seen in the BrO and HCHO daily level 3 product. Total



column $SO_2$ observations from GOME-2C are in general 0.5 DU lower than GOME-2A and B, resulting a slightly negative global average. Higher global average of $SO_2$ observed by GOME-2A and B is related to the extreme values taken with high solar zenith angle thus low signal to noise ratio (see Figure 4 and 5), while this effect is much less significant for GOME-2C. The overall bias and root mean square of error among the GOME-2 sensors for each product are summarized in Table 3.

**Table 3.** Bias and root mean square error of trace gas columns among the three GOME-2 sensors.

| Species (unit) | GOME-2B - GOME-2A[a] | | GOME-2C - GOME-2A[b] | | GOME-2C - GOME-2B[b] | |
|---|---|---|---|---|---|---|
| | Bias | RMSE | Bias | RMSE | Bias | RMSE |
| Total $O_3$ (DU) | $0.22 \pm 2.24$ | $5.13 \pm 1.52$ | $3.36 \pm 3.68$ | $7.41 \pm 2.52$ | $2.29 \pm 0.81$ | $4.60 \pm 1.00$ |
| Total $NO_2$ ($\times 10^{13}$ molec cm$^{-2}$) | $-2.35 \pm 6.31$ | $14.54 \pm 2.17$ | $12.05 \pm 7.56$ | $18.91 \pm 5.79$ | $12.70 \pm 3.84$ | $16.33 \pm 2.83$ |
| Tropo $NO_2$ ($\times 10^{13}$ molec cm$^{-2}$) | $0.69 \pm 2.94$ | $63.38 \pm 23.37$ | $-15.96 \pm 4.93$ | $77.22 \pm 11.92$ | $-14.86 \pm 3.59$ | $67.05 \pm 38.71$ |
| TCWV (kg m$^{-2}$) | $-0.14 \pm 0.36$ | $3.15 \pm 0.34$ | $-0.93 \pm 0.22$ | $3.35 \pm 0.42$ | $-0.52 \pm 0.09$ | $2.32 \pm 0.30$ |
| Total BrO ($\times 10^{12}$ molec cm$^{-2}$) | $-1.41 \pm 1.25$ | $5.34 \pm 1.03$ | $0.52 \pm 1.45$ | $6.22 \pm 0.59$ | $1.02 \pm 0.40$ | $3.37 \pm 0.30$ |
| Total HCHO ($\times 10^{15}$ molec cm$^{-2}$) | $1.54 \pm 0.41$ | $8.24 \pm 2.19$ | $1.89 \pm 0.54$ | $11.00 \pm 2.11$ | $-0.08 \pm 0.28$ | $5.68 \pm 0.55$ |
| Total $SO_2$ (DU) | $0.06 \pm 0.13$ | $1.21 \pm 0.46$ | $-0.53 \pm 0.34$ | $2.20 \pm 0.51$ | $-0.56 \pm 0.14$ | $2.08 \pm 0.51$ |

[a] for period from 2013 to 2021

[b] for period from 2019 to 2021

## 5.1.2 Zonal average

Each GOME-2 monthly averaged level 3 product derived from all three sensors is sorted by latitude and plotted in Figure 7. All three GOME-2 sensors show consistent zonal and seasonal $O_3$ patterns. Higher $O_3$ columns are observed over high latitudes, and lower values are found over the tropics.

Total column $O_3$ over the Arctic shows a peak in February to March and a minimum in August to October, while Antarctica displays a reverted seasonal pattern. Both total and tropospheric column $NO_2$ from all three GOME-2 sensors show good zonal and seasonal consistency. Elevated total column $NO_2$ are observed in the polar regions during the warm months. This seasonal pattern is attributed to the stratospheric variation of $NO_2$. Compared to total column $NO_2$, tropopsheric column $NO_2$ shows a very different zonal and seasonal pattern. Tropospheric $NO_2$ is mostly concentrated at the mid-latitudes of the northern hemisphere. It is because most of the population are living in this part of the world, thus higher emissions occur at this latitude band. Tropospheric $NO_2$ at mid-latitudes also shows a seasonal pattern with higher values over winter, which is related to higher energy consumption and longer atmospheric lifetime of $NO_2$ during the cold months. A significant increasing trend of tropospheric $NO_2$ can be observed by GOME-2A and B over the sub-tropics and mid-latitudes of the southern hemisphere in the recent years (see Figure 7g and h). GOME-2C observed a much less significant enhancement of tropospheric $NO_2$ in the southern hemisphere, which leads to lower global average tropospheric $NO_2$ measured by GOME-2C. This discrepancy is likely related to the difference in retrieval wavelength and the subsequent stratosphere and troposphere separation process.



**Figure 7.** Monthly zonal average of total column $O_3$ (1st row), total column $NO_2$ (2nd row), tropospheric column $NO_2$ (3th row), total column water vapour (4th row), total column BrO (5th row), (total column HCHO (6th row) and total column $SO_2$ (7th row). Data from GOME-2A (1st column from the left), GOME-2B (2nd column from the left) and (3th column from the left) are shown.

Total column water vapour observations from all three GOME-2 sensors show consistent zonal and seasonal patterns, with higher values in the tropic and lower at high latitudes. Total column water vapour is also higher during the warm months of the corresponding hemisphere.



All three GOME-2 sensors also show very similar zonal and seasonal patterns of total column BrO. However, GOME-2A total column BrO observations from 2014 to 2019 are slightly higher than that of GOME-2B at all latitude bands and results a small bias of $1.41 \times 10^{12}$ molec cm$^{-2}$. However, when we look into the data from 2020 to 2021, the bias is smaller and result a smaller bias of $0.52 \times 10^{12}$ molec cm$^{-2}$ with GOME-2C observations.

Total column HCHO from all three GOME-2 sensors show higher values over tropics and sub-tropics, while lower values appear at higher latitudes. Both GOME-2A and B measurements show a significant decreasing trend of HCHO in the southern hemisphere. However, GOME-2A measurements are significantly lower than GOME-2B and C, resulting a bias of -1.54 and -1.89 $\times 10^{15}$ molec cm$^{-2}$ when compared to GOME-2B and C observations. The discrepancy is related to the underestimation over HCHO rich regions, e.g., Amazon, Southeast Asia and Australia (see Figure 5).

Total column SO$_2$ observations from all three GOME-2 sensors show very low SO$_2$ levels (very close to 0) around the globe as expected. However, GOME-2A and B measurements show significantly higher noise for measurement with high solar zenith angle and result a small overestimation under these extreme observation geometries, while this effect are much less significant for GOME-2C. Therefore, GOME-2C observations are in general about 0.5 DU lower than GOME-2A and B.

## 5.2    Comparison to ground-based observations

In this section, each GOME-2 level 3 products are compared to the corresponding reference ground-based observations. We are looking into the scatter plot, histogram of the differences, and sort the differences/bias by year, latitude band or measurement site as box plot and time series between GOME-2 and reference data sets to investigate the systematic bias/error.

### 5.2.1    Total column ozone

Daily and monthly GOME-2 level 3 total column ozone are compared to the co-located Brewer observations. Figure 8 shows
the density scatter plots for the comparison of total column ozone between GOME-2 and ground-based Brewer observations. Comparisons of GOME-2A, B and C data are shown in Figure 8a, b and c, respectively. Monthly data are also shown as black dots. Histograms of the differences between GOME-2 and Brewer observations are shown in Figure 8d. Scatter plots show that GOME-2 monthly data is well in line with the daily data. And the agreement between GOME-2 and Brewer is in general very good with Pearson correlation coefficient ($R$) of $\sim$0.96 for all three GOME-2 sensors. The slopes of the total least squares
regression for the comparisons of all three instruments are very close to 1 (1.03 for GOME-2A, 1.01 for GOME-2B, and 0.99 for GOME-2C). The offsets of the total least squares regression range between -5.1 to 4.3 DU. In general, the GOME-2 data sets show a small positive bias of 2.3 to 3.5 DU compared to Brewer observations with standard deviation of 13.9 to 14.7 DU. The bias between all three GOME-2 sensors and ground-based Brewer observations is below 1 % which is within the uncertainty of Brewer measurements (Kerr et al., 1988) and fulfils the product requirements.

Figure 9 shows box plots of the differences of total column ozone between GOME-2 level 3 product and co-located Brewer measurements. GOME-2 data is sorted by the measurement year (Figure 9a) and latitude band (Figure 9b). The box plot for the southern hemisphere is mostly empty due to insufficient number of ground-based observations. The mean difference between GOME-2 and Brewer observations are within 5 DU for most of the years. However, we observed that there are years with

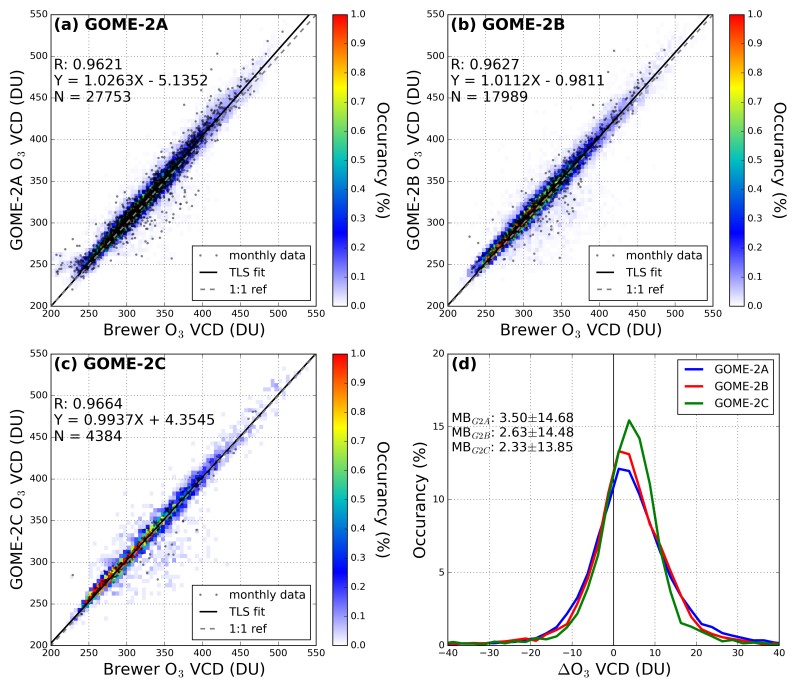

**Figure 8.** Comparison of daily and monthly total column $O_3$ measured by the ground-based Brewer instruments to (a) GOME-2A, (b) GOME-2B and (c) GOME-2C. Histograms of the differences of total column $O_3$ between GOME-2 and Brewer observations are shown in (d). Co-located daily and monthly averaged data are used in the comparison. Total least squares regression is based on daily data.

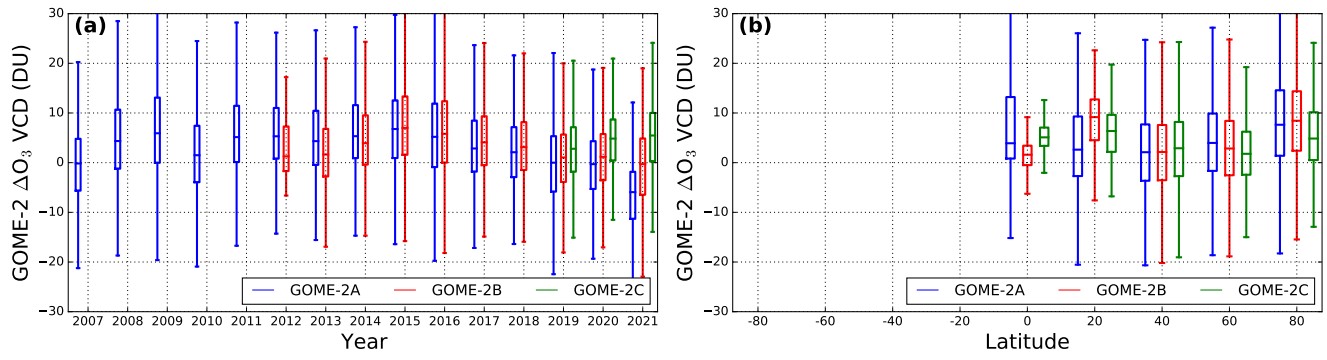

**Figure 9.** Comparison of total column $O_3$ between ground-based Brewer instruments and GOME-2 observations. Data are sorted by year in (a), and latitude band in (b).

positive bias while some years with negative bias. This is mostly related to the availability of ground-based data at different measurement sites. As some sites are bias high/low, and it will affect the statistic if they are not available for some years. On

the other hand, the latitude dependent analysis shows that GOME-2 observations is consistently higher than the ground-based Brewer measurements in the Northern Hemisphere and result a positive bias of 2.3 to 3.5 DU on average. In addition, GOME-2C observations are about 2 - 3 DU higher than GOME-2A and B, which is likely relate to the instrumental issues which has been mentioned in Section 5.1.1.

## 5.2.2 Total column $NO_2$

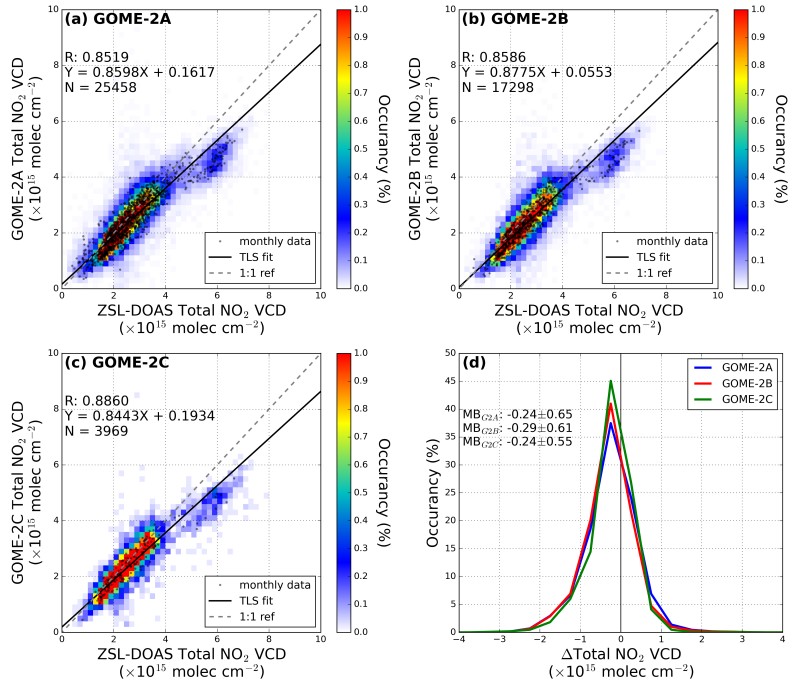

**Figure 10.** Comparison of daily and monthly total column $NO_2$ measured by the ground-based ZSL-DOAS to (a) GOME-2A, (b) GOME-2B and (c) GOME-2C. Histograms of the difference of total column $NO_2$ between GOME-2 and ZSL-DOAS observations are shown in (d). Co-located daily and monthly averaged data are used in the comparison. Total least squares regression is based on daily data.

Daily and monthly GOME-2 level 3 total column $NO_2$ are compared to the co-located ZSL-DOAS observations. Figure 10 shows the density scatter plots for the comparison of total column $NO_2$ between GOME-2 and ground-based ZSL-DOAS observations. Comparisons of GOME-2A, B and C data are shown in Figure 10a, b and c, respectively. Monthly data are also shown as black dots. Histograms of the differences between GOME-2 and ZSL-DOAS observations are shown in Figure 10d. Scatter plots show that GOME-2 monthly data is well in line with the daily data. GOME-2 level 3 total column $NO_2$ is in general agree well with ZSL-DOAS observations with Pearson correlation coefficient ($R$) of 0.85 to 0.88. However, GOME-2 observations are in general slightly lower than ZSL-DOAS observations. The slopes of the total least squares fit for the comparisons of all three instruments vary from 0.84 to 0.88 with offset ranging from 0.05 - 0.19 $\times 10^{15}$ molec cm$^{-2}$. Overall, the GOME-2 level

3 total $NO_2$ products are biased low by $0.24 - 0.29 \times 10^{15}$ molec cm$^{-2}$ compared to ground-based ZSL-DOAS measurements. Considering that the uncertainty of satellite and ground-based measurements is about 10 %, the agreement between the two dataset is very satisfactory.

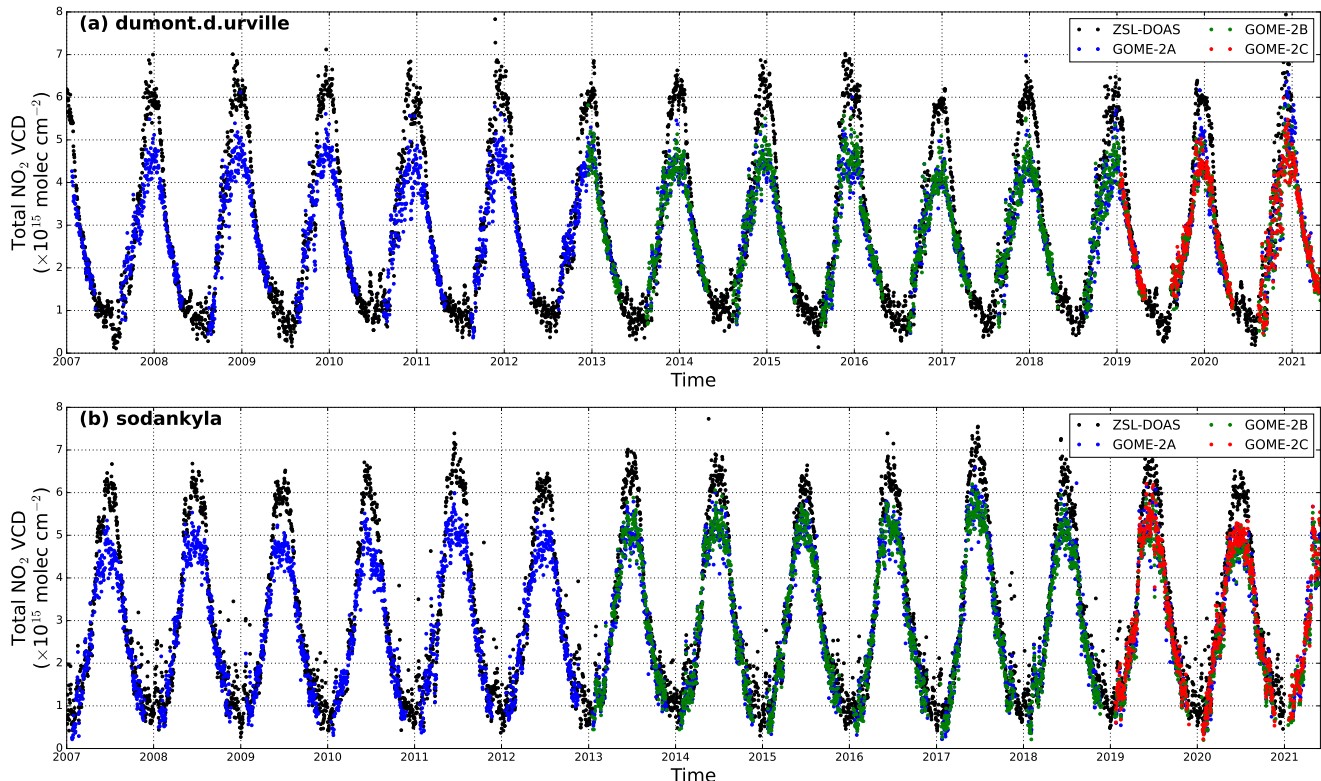

**Figure 11.** Time series of total column $NO_2$ measured by GOME-2A (blue), GOME-2B (green), GOME-2C (red) and ZSL-DOAS (black). Observations over (a) Dumont d'Urville, Antarctica and (b) Sodankylä, Finland are shown.

The scatter plots for all three instruments show a two clusters characteristic. The major cluster of total column $NO_2$ below $4 \times 10^{15}$ molec cm$^{-2}$ shows very good agreement between GOME-2 and ZSL-DOAS observations. The minor cluster at $5 - 6 \times 10^{15}$ molec cm$^{-2}$ shows significant underestimation of $NO_2$ column by $0.5 - 1.0 \times 10^{15}$ molec cm$^{-2}$ which is related to the measurement over Polar regions. Figure 11 shows the time series of total column $NO_2$ measured at Dumont d'Urville, Antarctica and Sodankylä, Finland. We observed that the total column $NO_2$ measured by GOME-2 is significantly lower than the ground-based ZSL-DOAS observations during summer months. This is because of the multiple overpasses over Polar Regions during summertime. Therefore, GOME-2 level 3 data represents the real "daily average" while ZSL-DOAS only capture the morning values. Due to the diurnal variation of $NO_2$, it is expected that ZSL-DOAS measurements in the morning



is higher than the daily averages. If we do not consider these two stations in the analysis, the minor cluster in the scatter plots would be removed. In addition, the underestimation would reduce to $0.13 - 0.21 \times 10^{15}$ molec cm$^{-2}$.

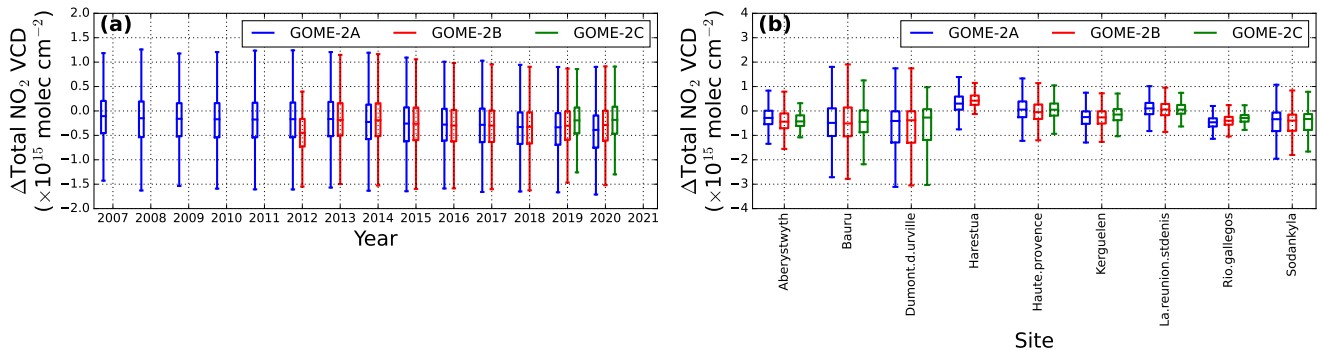

**Figure 12.** Comparison of total column NO$_2$ between ground-based ZSL-DOAS and GOME-2 observations. Data are sorted by year in (a), and measurement site in (b).

Figure 12 shows box plots of the differences of total column NO$_2$ between GOME-2 level 3 product and co-located ZSL-DOAS measurements. Data are sorted by the measurement year (Figure 12a) and measurement site (Figure 12b). The mean differences between GOME-2 and ZSL-DOAS observations are within $0.3 \times 10^{15}$ molec cm$^{-2}$ for most of the years and this bias does not show significant temporal variation. Box plots for each measurement site show significant negative bias for some sites, i.e., Dumont d'Urville and Sodankylä. The reason of the negative bias has been explained above.

### 5.2.3 Tropospheric column NO$_2$

Daily and monthly GOME-2 level 3 tropospheric column NO$_2$ are compared to the co-located MAX-DOAS observations. Figure 13 shows the density scatter plots for the comparison of tropospheric column NO$_2$ between GOME-2 and ground-based MAX-DOAS observations. Comparisons of GOME-2A, B and C data are shown in Figure 13a, b and c, respectively. Monthly data are also shown as black dots. Histograms of the differences between GOME-2 and MAX-DOAS observations are shown in Figure 13d. GOME-2 monthly tropospheric NO$_2$ data is consistent with the daily data, and daily data shows satisfactory correlation with ground-based MAX-DOAS observations with Pearson correlation coefficient ($R$) in a range of 0.68 to 0.75. However, GOME-2 tropospheric column NO$_2$ are in general ∼30 % lower than MAX-DOAS observations. The slopes of the total least squares fit for the comparisons of all three instruments vary from 0.61 to 0.74 with offset ranging from -1.03 to $0.18 \times 10^{15}$ molec cm$^{-2}$. GOME-2 level 3 tropospheric NO$_2$ products on average show a negative bias of $3.38 - 4.14 \times 10^{15}$ molec cm$^{-2}$. The underestimation is mainly related to the a-priori assign too low NO$_2$ concentration at the lower troposphere and spatial averaging effect over large satellite pixel. Previous study shows that using better a-priori vertical profile in GOME-2 retrieval reduces the underestimation of GOME-2 measurement by 15 - 20 % (Liu et al., 2019). The spatial averaging effect has also been estimated to result an underestimation of 15 - 25 % in tropospheric column NO$_2$ over pollution

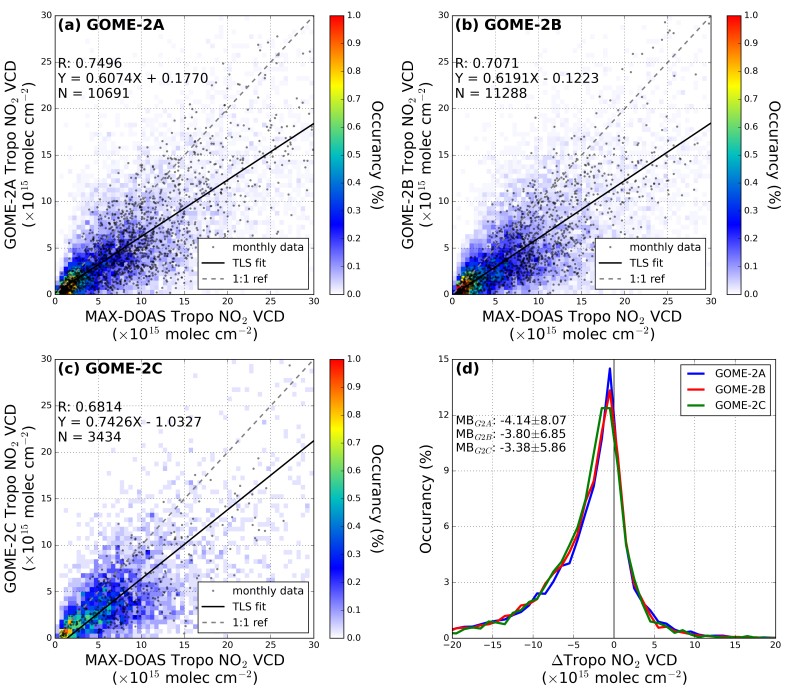

**Figure 13.** Comparison of daily and monthly tropospheric column $NO_2$ measured by the ground-based MAX-DOAS to (a) GOME-2A, (b) GOME-2B and (c) GOME-2C. Histograms of the difference of tropospheric column $NO_2$ between GOME-2 and MAX-DOAS are shown in (d). Co-located daily and monthly averaged data are used in the comparison. Total least squares regression is based on daily data.

hotspots (Chen et al., 2009; Ma et al., 2013; Chan et al., 2020b; Pinardi et al., 2020a). Considering that the sensitivity difference between satellite and ground-based MAX-DOAS measurements and the spatial averaging effect of large satellite footprint, the agreement between the two dataset is very satisfactory.

Figure 14 shows box plots of the differences of tropospheric column $NO_2$ between GOME-2 level 3 product and co-located
5   MAX-DOAS measurements. Data is sorted by the measurement year (Figure 14a) and measurement site (Figure 14b). The mean differences between GOME-2 and MAX-DOAS observations are $\sim 3 \times 10^{15}$ molec cm$^{-2}$ for most of the years and this bias do not show significant temporal variation. Box plots for each measurement site show significant negative bias for some polluted sites, i.e., Beijing (China), Thessaloniki (Greece) and Yokosuka (Japan). The reason of the negative bias has been explained above. The underestimation is significantly reduced over rural areas, e.g., Cape Hedo (Japan), Cabauw (Netherlands)
10  and Phimai (Thailand). These results are in line with the level 2 data that GOME-2 in general underestimates tropospheric column $NO_2$ over polluted areas.

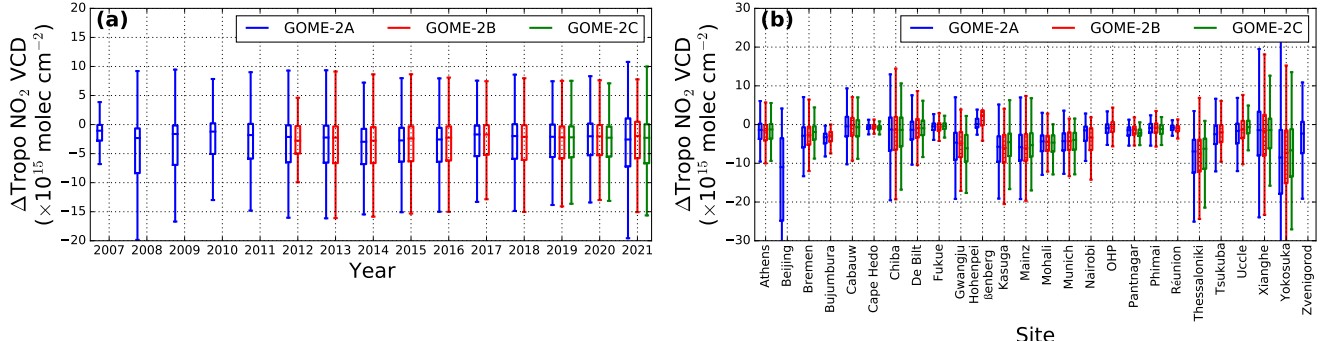

**Figure 14.** Comparison of tropospheric column $NO_2$ between ground-based MAX-DOAS and GOME-2 observations. Data are sorted by year in (a), and measurement site in (b).

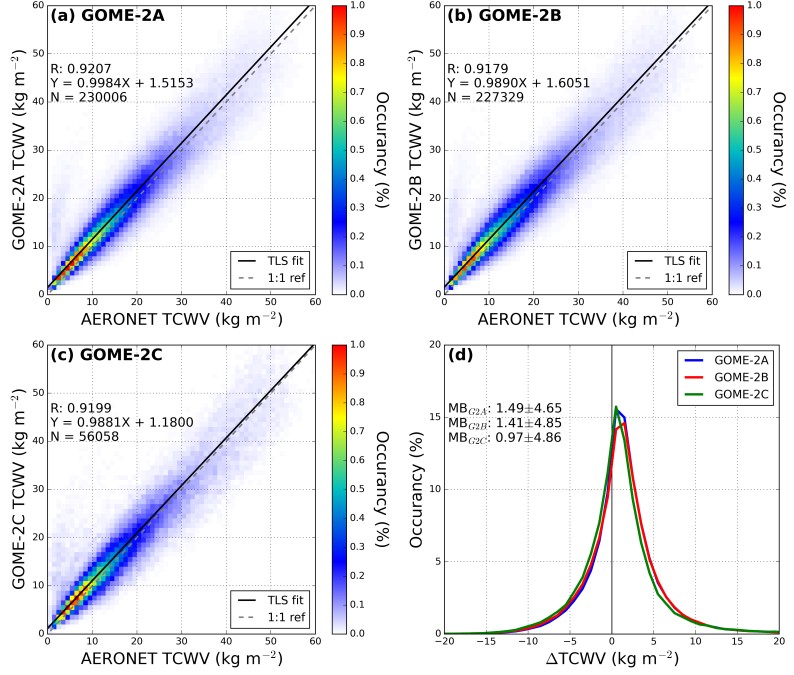

**Figure 15.** Comparison of daily total column water vapour measured by the sun photometer to (a) GOME-2A, (b) GOME-2B and (c) GOME-2C. Histograms of the difference between GOME-2 and sun-photometer are shown in (d). Co-located daily averaged data are used in the comparison.

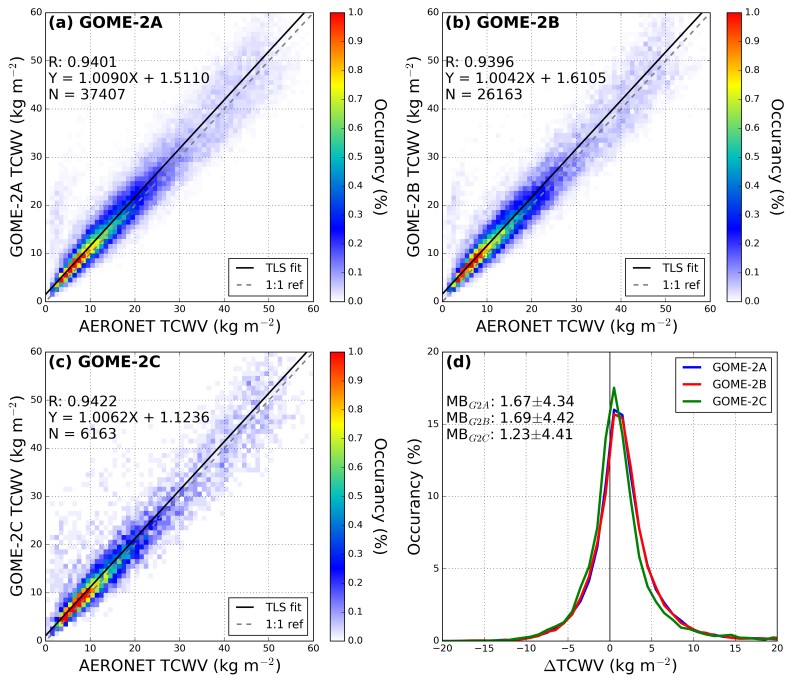

**Figure 16.** Comparison of monthly total column water vapour measured by the sun photometer to (a) GOME-2A, (b) GOME-2B and (c) GOME-2C. Histograms of the difference between GOME-2 and sun-photometer are shown in (d). Co-located monthly averaged data are used in the comparison.

### 5.2.4 Total column water vapour

Daily GOME-2 level 3 total column water vapour are compared to the co-located sun-photometer observations. Figure 15 shows the density scatter plots for the comparison of total column water vapour column between GOME-2 and ground-based sun-photometer observations. Comparisons of GOME-2A, B and C data are shown in Figure 14a, b and c, respectively. His-5 tograms of the differences between GOME-2 and MAX-DOAS observations are shown in Figure 15d. Similar plots for monthly comparison are shown in Figure 16. GOME-2 monthly total column water vapour data is in general consistent with the daily data. GOME-2 daily observations are in good agreement with sun-photometer observations, with Pearson correlation coefficient ($R$) of ~0.92 for all three instruments. Monthly comparison shows higher correlation coefficient ($R$) of ~0.94. The slopes of least squares regression lines of daily comparison for all three GOME-2 sensors are very close to 1, while a small offset of 10 1.2 - 1.6 kg m$^{-3}$ is observed. Monthly comparison shows similar characteristic with the slope of regression close to 1 and offset of 1.1 - 1.6 kg m$^{-3}$. GOME-2 level 3 total column water vapour in general show a positive bias of 1.0 - 1.7 kg m$^{-3}$. Considering that sun-photometer measurements are in general underestimating total column water vapour by 6- % (Pérez-Ramírez et al., 2014) the positive bias of 1.0 - 1.7 kg m$^{-3}$ is reasonable.

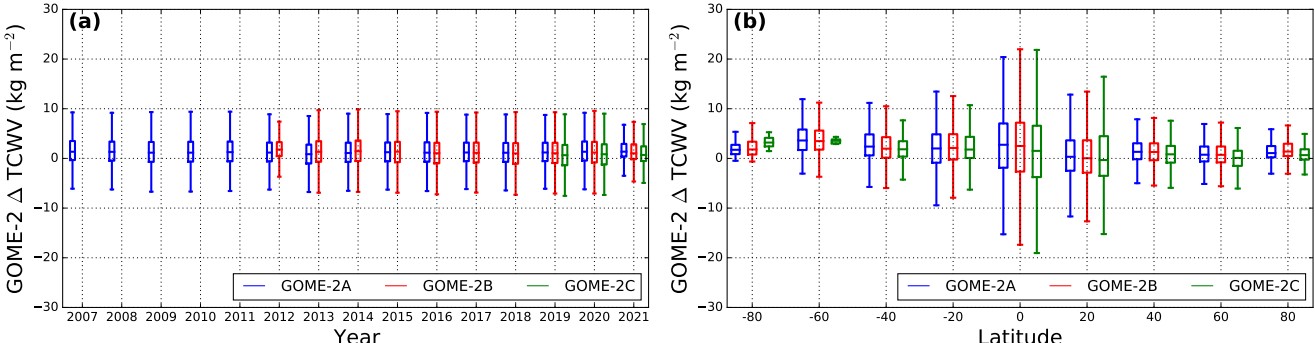

**Figure 17.** Comparison of TCWV between sun photometer and GOME-2 observations. Data are sorted by year in (a), and latitude in (b).

Figure 17 shows box plots of the statistic of the differences of total column water vapour between GOME-2 level 3 product and co-located sun-photometer measurements. Data is sorted by the measurement year (Figure 17a) and latitude band (Figure 17b). The bias between GOME-2 and sun-photometer observations is consistently at level of $1$ - $2\,\mathrm{kg\,m^{-3}}$ throughout the entire measurement period. The latitude dependency analysis shows larger variations in the tropics, while the variations are
much smaller at higher latitudes. The absolute differences for measurements over Polar Regions are slightly higher. This is mainly due to multiple overpasses over Polar Regions during summer months and resulting temporal mismatch.

### 5.2.5    Total column BrO

Co-located daily and monthly GOME-2 level 3 total column BrO are compared to ZSL-DOAS observations at Harestua, Norway. Figure 18 shows the density scatter plots for the comparison of total column BrO between GOME-2 and ZSL-DOAS
observations. Comparisons of GOME-2A, B and C data are shown in Figure 18a, b and c, respectively. Monthly data are also shown as black dots. Histograms of the differences between GOME-2 and ZSL-DOAS observations are shown in Figure 18d. We can see from the scatter plots that both GOME-2 and ZSL-DOAS BrO measurements are quite noisy, it is mainly due to the low absorption of BrO and thus low signal to noise ratio. Both daily and monthly GOME-2 level 3 data show quite good agreement with the ZSL-DOAS observations, with Pearson correlation coefficient ($R$) ranging from 0.64 to 0.74. In general,
GOME-2 observations are underestimating BrO column by $7.0$ - $10.2 \times 10^{12}\,\mathrm{molec\,cm^{-2}}$.

Figure 19 shows the time series of total column BrO measured at Harestua, Norway. Measurements from all three GOME-2 sensors show similar temporal variation trend with higher BrO level during summer and lower in winter which agrees with the ZSL-DOAS observations. However, GOME-2 observations are about $5$ - $10 \times 10^{12}\,\mathrm{molec\,cm^{-2}}$ lower than the ZSL-DOAS data. This underestimation has also been reported in the level 2 validation report (Theys et al., 2015). Considering that the
ZSL-DOAS data have been empirically corrected for the offset caused by instrumental effect, the agreement between GOME-2 and ZSL-DOAS is deemed very satisfactory.



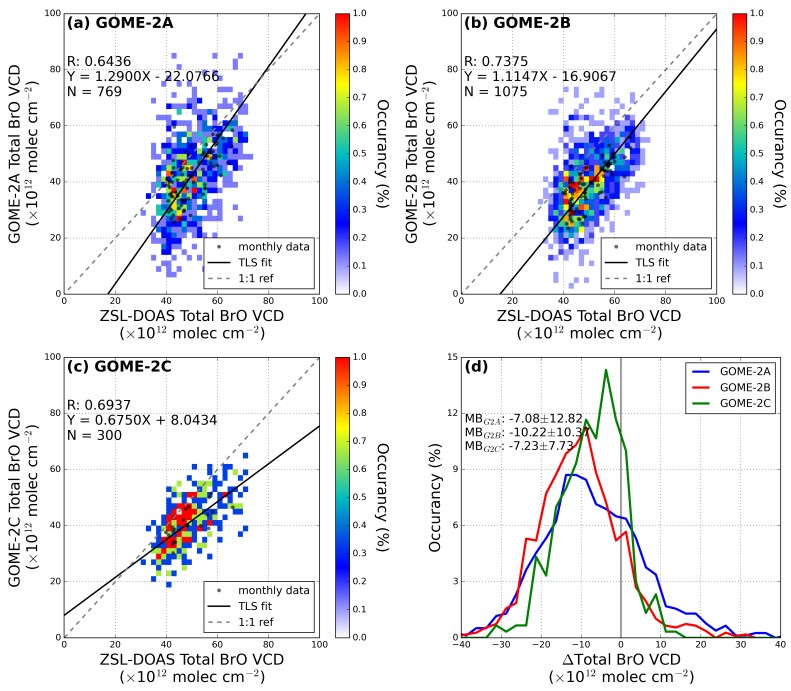

**Figure 18.** Comparison of daily and monthly total column BrO measured by the ground-based ZSL-DOAS at Harestua, Norway to (a) GOME-2A, (b) GOME-2B and (c) GOME-2C. Histograms of the difference of total column BrO between GOME-2 and MAX-DOAS observations are shown in (d). Co-located daily and monthly averaged data are used in the comparison. Total least squares regression is based on daily data.

### 5.2.6 Total column HCHO

Daily and monthly GOME-2 level 3 total column HCHO are compared to the co-located MAX-DOAS observations. Figure 20 shows the density scatter plots for the comparison of total column HCHO between GOME-2 and ground-based MAX-DOAS observations. Comparisons of GOME-2A, B and C data are shown in Figure 20a, b and c, respectively. Monthly data are also
5    shown as black dots. Histograms of the differences between GOME-2 and MAX-DOAS observations are shown in Figure 20d. We can see from the scatter plots that both GOME-2 and MAX-DOAS HCHO measurements are quite noisy, it is mainly due to the low absorption of HCHO and thus low signal to noise ratio. However, when we look at the monthly averages, the GOME-2 level 3 data in general agrees with the ground-based MAX-DOAS observations. The Pearson correlation coefficient ($R$) between monthly GOME-2 and MAX-DOAS data ranges from 0.68 to 0.78. However, GOME-2 observations are in general
10    underestimating total column HCHO by $20 \text{-} 25 \%$. The slope of the total least squares regression line for the comparisons of all three instruments varies from 0.74 to 0.81 with offset ranging from -1.61 to $-1.14 \times 10^{15}$ molec cm$^{-2}$. GOME-2 level 3 total HCHO products on average show a small bias of -0.75 to $1.92 \times 10^{15}$ molec cm$^{-2}$ with standard deviation of 8.8 up to

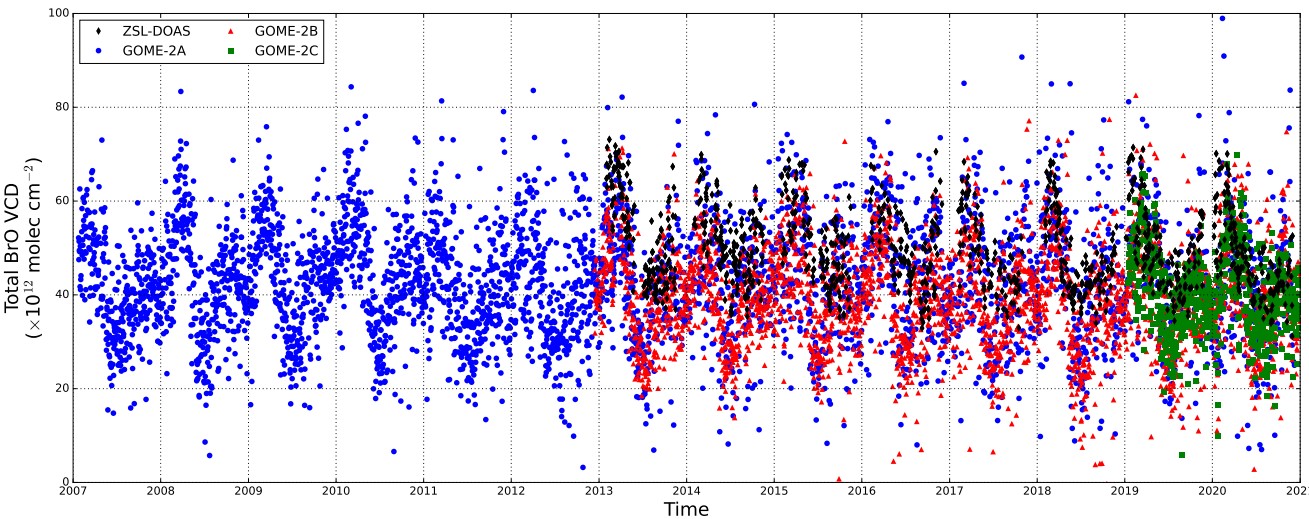

**Figure 19.** Time series of total column BrO measured by GOME-2A (blue), GOME-2B (green), GOME-2C (red) and ZSL-DOAS (black) at Harestua, Norway.

$11.4 \times 10^{15}$ molec cm$^{-2}$. The underestimation is partly related to the a-prior profile used in GOME-2 retrieval and difference of sensitivity between satellite and ground-based observations. The underestimation of level 3 product is in line with the level 2 product. Previous studies shows that the negative bias is significantly improved when MAX-DOAS profile is used for satellite column retrieval (De Smedt et al., 2015a, b).

Figure 21 shows box plots of the statistic of the differences of total column HCHO between GOME-2 level 3 product and co-located MAX-DOAS measurements. Data is sorted by the measurement year (Figure 21a) and measurement site (Figure 21b). The mean differences between GOME-2 and MAX-DOAS observations are 1 - 2 $\times 10^{15}$ molec cm$^{-2}$ for most of the years and do not show significant temporal variation. Box plots for each measurement site show that GOME-2 significantly underestimated HCHO column over polluted areas, i.e., Mexico City (Mexico) and Xianghe (China). The underestimation is related to

the difference in sensitivity and this effect has been reported in previous level 2 validation studies for GOME-2 (De Smedt et al., 2015b; Pinardi et al., 2020b) as well as for other satellites (Chan et al., 2020b; De Smedt et al., 2021).

### 5.2.7   Total column SO$_2$

Co-located daily and monthly GOME-2 level 3 total column SO$_2$ are compared to Pandora observations at Mexico City. Figure 22 shows the scatter plots for the comparison of total column SO$_2$ between GOME-2 and Pandora observations. Com-

parisons of GOME-2A, B and C data are shown in Figure 22a, b and c, respectively. Monthly data are also shown as black dots. Histograms of the differences between GOME-2 and Pandora observations are shown in Figure 22d. Due to the low





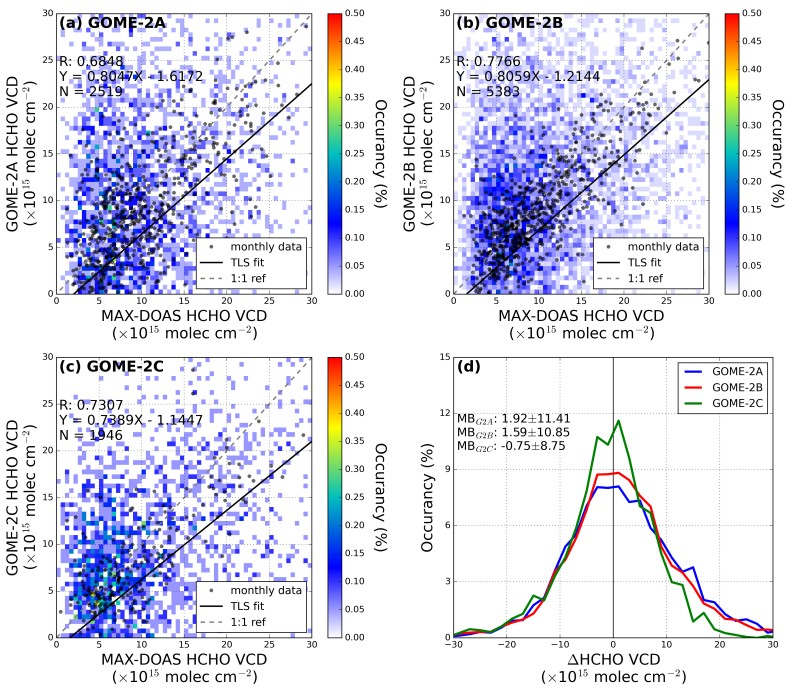

**Figure 20.** Comparison of daily and monthly total column HCHO measured by the ground-based MAX-DOAS to (a) GOME-2A, (b) GOME-2B and (c) GOME-2C. Histograms of the difference of total column HCHO between GOME-2 and MAX-DOAS are shown in (d). Co-located daily and monthly averaged data are used in the comparison. Total least squares regression is based on daily data.

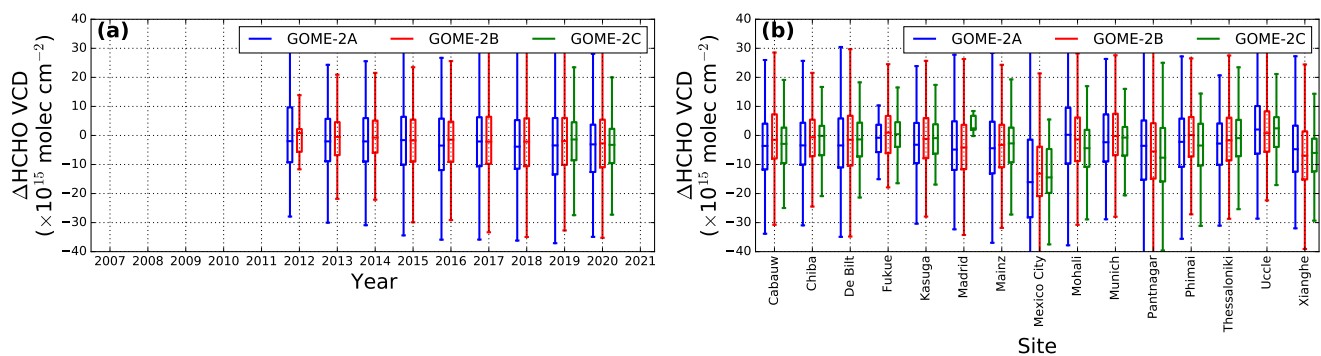

**Figure 21.** Comparison of total column HCHO between ground-based MAX-DOAS and GOME-2 observations. Data are sorted by year in (a), and measurement site in (b).

absorption and abundancy of SO$_2$, both GOME-2 and Pandora measurements are quite noisy. Histogram shows that GOME-2 underestimated total column SO$_2$ by 0.25 to 0.48 DU.



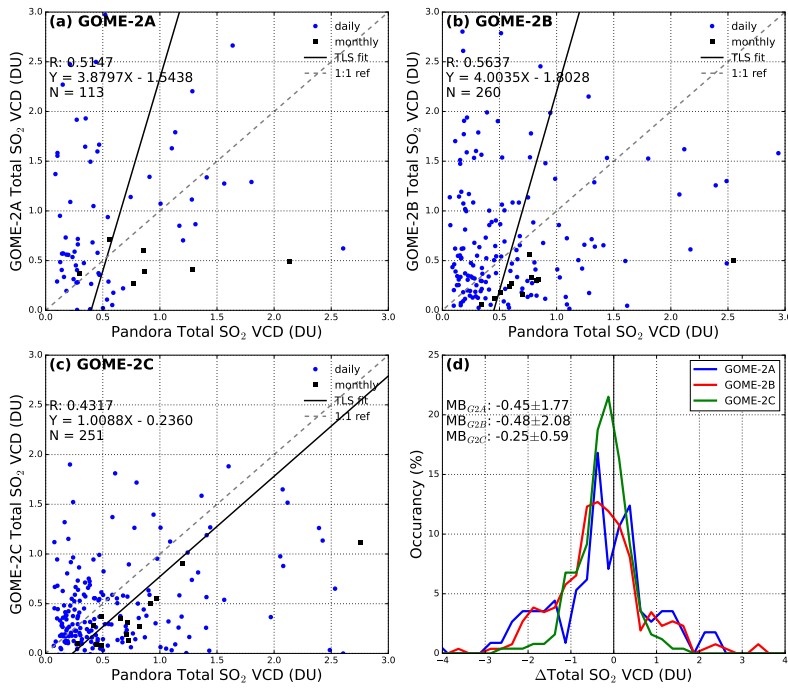

**Figure 22.** Comparison of daily and monthly total column SO$_2$ measured by Pandora instrument in Mexico City to (a) GOME-2A, (b) GOME-2B and (c) GOME-2C. Histograms of the difference of total column SO$_2$ between GOME-2 and Pandora observations are shown in (d). Co-located daily and monthly averaged data are used in the comparison. Total least squares regression is based on daily data.

Figure 23 shows the time series of total column SO$_2$ measured at Mexico City. All three GOME-2 sensors show similar SO$_2$ columns. The overall averages are very close to zero and do not show any significant trend. Due to the low abundancy of SO$_2$ and low signal to noise ratio, there are considerable number of negative values. On the other hand, due to better signal to noise ratio, only very few negative values measured by Pandora. Considering the measurement noise of GOME-2, the agreement between GOME-2 and Pandora datasets is reasonable.

## 6  Summary

We presented the new GOME-2 daily and monthly level 3 products which include total column O$_3$, total and tropospheric column NO$_2$, total column water vapour, total column BrO, total column HCHO and total column SO$_2$. Details of the algorithm for level 2 to level 3 processing as well as the selection of appropriate spatial resolution for the level 3 products are shown. Verification and validation of each GOME-2 level 3 product are achieved by investigating the consistency among the three GOME-2 sensors and comparison to ground-based reference measurements.

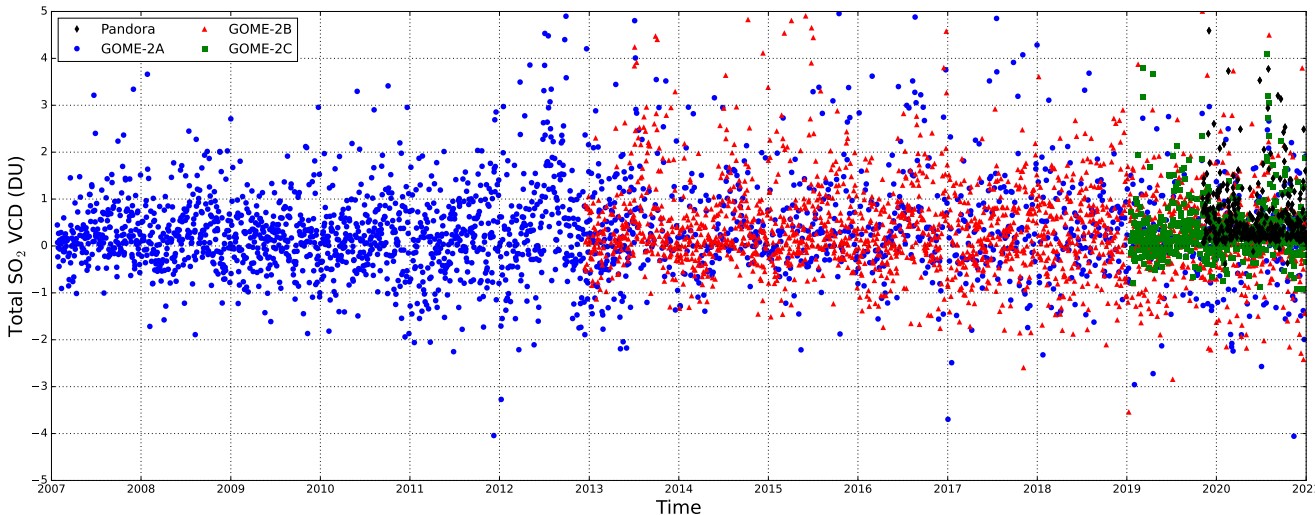

**Figure 23.** Time series of total column SO$_2$ measured by GOME-2A (blue), GOME-2B (green), GOME-2C (red) and Pandora (black) at Mexico City.

The overlapping area weighting method is used for level 2 to level 3 processing. The spatial resolution of the GOME-2 level 3 products is selected based on sensitivity study. The consistency among three GOME-2 sensors is investigated through time series of global averages, zonal averages, and bias. Finally, the accuracy of the level 3 products is validated through the comparison to ground-based observations.

For the selection of appropriate spatial resolution of the GOME-2 level 3 data, we have re-sampled GOME-2 level 2 data onto various spatial resolutions, i.e., $0.1° \times 0.1°$, $0.25° \times 0.25°$ and $0.5° \times 0.5°$ and compared to the original level 2 data. All datasets show very similar spatial structures and the absolute values are consistency with the level 2 products. As expected, level 3 data sampled at higher spatial resolution (i.e., $0.1° \times 0.1°$) better preserved the original GOME-2 instrument footprint. However, lower resolution of $0.25° \times 0.25°$ also preserves the spatial pattern of fast varying tropospheric species, i.e., NO$_2$,

reasonably well. While a rather strong smoothing/averaging effect is observed from data gridded with lower spatial resolution (i.e., $0.5° \times 0.5°$). Therefore, we concluded that the spatial resolution of $0.25° \times 0.25°$ is sufficient and appropriate for GOME-2 level 3 products.

The consistency of level 3 product among the three GOME-2 sensors are investigated. Global average time series plots show that total column ozone and water vapour products from all GOME-2 sensors are consistent, with only a small bias of

up to 3 DU ($<1\,\%$) for ozone, and $0.9\,\mathrm{kg\,m^{-2}}$ ($<5\,\%$) for water vapour. For total and tropospheric column NO$_2$ products, GOME-2A and B measurements are consistent with each other, while GOME-2C data show significant discrepancy compared to the other two sensors. This is mainly due to the differences in processor versions (GDP 4.8 for GOME-2A & B and GDP 4.9 for GOME-2C) and spectral fitting band of NO$_2$. BrO observations from GOME-2B in general show a negative bias of





1.0 - $1.5 \times 10^{12}$ molec cm$^{-2}$ compared to GOME-2A and C. GOME-2A HCHO columns are 1.5 - $1.9 \times 10^{15}$ molec cm$^{-2}$ lower than GOME-2B and C measurements. This is due to the underestimation over Amazon, Southeast Asia, and Australia. Total column SO$_2$ observations from GOME-2C are on average 0.5 DU lower than GOME-2A and B, resulting a slightly negative global average. Slightly higher global average of SO$_2$ measured by GOME-2A and B is related to the high values taken under

extreme viewing geometry, i.e., high solar zenith angle.

For comparison of co-located GOME-2 level 3 data to ground-based observations, we found in general good agreement and the results are consistent with previous level 2 validation studies. We summarized the statistical result in Table 4.

**Table 4.** Summary of the GOME-2 level 3 data comparison to ground-based measurements.

| GOME-2 Product | Reference Measurement | Correlation Coefficient ($R$) | | | Mean Bias | | |
|---|---|---|---|---|---|---|---|
| | | GOME-2A | GOME-2B | GOME-2C | GOME-2A | GOME-2B | GOME-2C |
| Total Column O$_3$ | Brewer | 0.96 | 0.96 | 0.97 | $3.5 \pm 14.7^{(a)}$ | $2.6 \pm 14.5^{(a)}$ | $2.3 \pm 13.9^{(a)}$ |
| Total Column NO$_2$ | ZSL-DOAS | 0.85 | 0.86 | 0.89 | $-0.24 \pm 0.65^{(b)}$ | $-0.29 \pm 0.61^{(b)}$ | $-0.24 \pm 0.55^{(b)}$ |
| Tropospheric Column NO$_2$ | MAX-DOAS | 0.75 | 0.71 | 0.68 | $-4.1 \pm 8.1^{(b)}$ | $-3.8 \pm 6.9^{(b)}$ | $-3.4 \pm 5.9^{(b)}$ |
| Total Column Water Vapour | Sun-photometer | 0.92 | 0.92 | 0.92 | $1.5 \pm 4.7^{(c)}$ | $1.4 \pm 4.9^{(c)}$ | $1.0 \pm 4.9^{(c)}$ |
| Total Column BrO | ZSL-DOAS | 0.64 | 0.74 | 0.69 | $7.1 \pm 12.8^{(d)}$ | $10.2 \pm 10.4^{(d)}$ | $7.2 \pm 7.7^{(d)}$ |
| Total Column HCHO | MAX-DOAS | 0.68 | 0.78 | 0.73 | $1.9 \pm 11.4^{(b)}$ | $1.6 \pm 10.9^{(b)}$ | $-0.8 \pm 8.8^{(b)}$ |
| Total Column SO$_2$ | Pandora | 0.51 | 0.56 | 0.43 | $0.45 \pm 1.8^{(a)}$ | $0.48 \pm 2.1^{(a)}$ | $2.5 \pm 0.6^{(a)}$ |

$^{(a)}$ unit in DU

$^{(b)}$ unit in $10^{15}$ molec cm$^{-2}$

$^{(c)}$ unit in kg m$^{-2}$

$^{(d)}$ unit in $10^{12}$ molec cm$^{-2}$

From the results above, we conclude that the daily and monthly GOME-2 level 3 products of total column O$_3$, total and tropospheric column NO$_2$, total column water vapour, total column BrO, total column HCHO and total column SO$_2$ for

GOME-2A, GOME-2B and GOME-2C are consistent and fulfil the product requirements.

## 7   Data availability

The GOME-2 level 3 products described in this paper is available to public through the DLR FTP server (ftp://ftp.dfd.dlr.de: /put/ACSAF/Level-3/) and will be transfer to AC SAF FTP-server (ftp://acsaf.eoc.dlr.de/) in due course.

*Author contributions.* KLC conceptualized the paper, devised the methodology, developed the algorithm and validated the data sets. KLC

and PV managed the project. KLC, KPH, RL, PH and DL provided supports on GOME-2 level 2 data. KLC, GP, MVR, FH, TW, VK, AB,





AP, HI, YK, HT, YC, KP, JC, AC, UF, AR, JM, NB, RH, CRC and MW provided ground-based reference data. KLC prepared the manuscript with contributions from all the co-authors.

*Competing interests.* The authors declare that they have no conflict of interest.

*Acknowledgements.* The work described in this paper was carried out within the framework of the European Organization for the Exploitation
5 of Meteorological Satellites (EUMETSAT) Satellite Application Facility on Atmospheric Composition Monitoring (AC SAF) Continuous Development and Operations Phase (CDOP-3) and Associated Scientist Project. We acknowledge EUMETSAT and AC SAF for the production of the GOME-2 level 2 and 3 data. We thank WOUDC, NDACC, NIDFORVAL and AERONET in harmonizing and managing the ground-based data sets used in this study. Work by Hitoshi Irie on ground-based MAX-DOAS observations data was supported the Environment Research and Technology Development Fund (JPMEERF20215005) of the Environmental Restoration and Conservation Agency of
10 Japan, JSPS KAKENHI (grant numbers JP20H04320, JP21K12227, JP22H03727, and JP22H05004), the JAXA 3rd research announcement on the Earth Observations (grant number 19RT000351), and the Virtual Laboratory (VL) project by the Ministry of Education, Culture, Sports, Science and Technology (MEXT), Japan.



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
