# Peer review of "Global Ozone Monitoring Experiment-2 (GOME-2) Daily and Monthly Level 3 Products of Atmospheric Trace Gas Columns"

_Earth System Science Data, 2022_

## Referee Comment (RC2)

Anonymous Reviewer

November 20, 2022

The paper by Chan et al. presents the GOME-2 Level-3 data of total column ozone, total and tropospheric column nitrogen dioxide, total column water vapour, total column bromine oxide, total column formaldehyde and total column sulphur dioxide. The topic fits well to the aims and scopes of ESSD. The manuscript is overall very well-written and has a clear structure. The described methods and validation results seem reasonable. I would favorably recommend a publication after the revised manuscript could

**1 Specific comments**

- In Introduction, it would be nicer to mention instruments like SCIAMACHY, OMI, and TROPOMI that measure atmospheric constituents within UV-VIS.

- As you mentioned that cloud parameters done by the OCRA/ROCINN algorithms, the cloud model CRB (Clouds-as-Reflecting-Boundaries) was used for GOME-2 A/B/C. Why the cloud model CAL (Clouds-As-Layers) wasn't used since CAL has been included in OCRA/ROCINN and (Loyola et al., 2018)? A new surface albedo climatology based on hyperspectral UV-VIS measurements has been introduced for TROPOMI (Loyola et al., 2020). Would it be also applied to GOME-2 data processing as well? Please extend the discussion in the manuscript.

- Page 2, Line 13-15: Please rewrite "Together with its successors . . . 25 years".

- Page 2, Line 25: Why "usually"? Any other format for expressing trace gases columns?

- Page 4, Table 1: Would it be possible to summarize the major differences between GDP 4.8 and GDP 4.9, except for different sensors?

- Page 21, Sect. 5.1.1: Different instrument characteristics like scan angle dependency and polarization sensitivity seem to have higher impact on UV species ($O_3$, BrO, HCHO) than VIS species ($NO_2$ and $H_2O$). Is that true?

- Page 23, Line 9: This sentence should be placed in the previous paragraph.

- Page 23, Line 17: *was* observed by . . .

- Page 25, Sect. 5.2.1: If available, could the authors provide relative differences for total ozone comparison between GOME-2 and ground-based measurements, which has been widely used in the total ozone validation?

- Page 39, Sect. 6: I suggest to add a paragraph to discuss the usefulness of GOME-2 Level-3 data. As compared to TROPOMI, OMI, and other data, would the authors still recommend it to the community?

**References**

Loyola, D. G., Gimeno García, S., Lutz, R., Argyrouli, A., Romahn, F., Spurr, R. J. D., et al. (2018). The operational cloud retrieval algorithms from TROPOMI on board Sentinel-5 Precursor. *Atmos. Meas. Tech.* 11, 409–427. doi:10.5194/amt-11-409-2018

Loyola, D. G., Xu, J., Heue, K.-P., and Zimmer, W. (2020). Applying FP_ILM to the retrieval of geometry-dependent effective lambertian equivalent reflectivity (GE_LER) daily maps from UVN satellite measurements. *Atmos. Meas. Tech.* 13, 985–999. doi:10.5194/amt-13-985-2020

---

## Referee Comment (RC3)

Review of **Global Ozone Monitoring Experiment-2 (GOME-2) Daily and Monthly Level 3 Products of Atmospheric Trace Gas Columns** by Ka Lok Chan et al.

This paper gives a full description of a new set of daily and monthly Level 3 products created from Level 2 Products (Total Column $O_3$, Total and Troposphere $NO_2$, Total $H_2O$, Total BrO, total HCHO, and total $SO_2$) estimated by using measurements from the Eumetsat GOME-2 instruments on the Metop platforms. It provides a clear description of the choices and statistical approaches used for the different species as well as validation versus ground-based measurements and the original Level 2 products. The paper includes summaries of the strengths and weaknesses for each component together with extensive citations of the literature.

There is always a loss of information in the creation of a Level 3 product in exchange for the filtering, weighting, and selection exercised by the developer. Is there a way to preserve or access the information on the a priori profiles and the viewing geometry for the Level 3 products? If not, then it becomes difficult to determine whether the column values are consistent with comparative profile measurements.

Given the differences in SZAs and local times is it worth separating out the ascending and descending measurements? Can the repeated measurements at high latitudes from ascending and descending portions of the orbits be used to investigate diurnal variations?

It should be possible to identify and flag many of the Level 1 spectra affected by charged particle events within the South Atlantic Anomaly. Is this information available from the Level 2 Products? While the expected effects are an increase in the recorded signal, they will produce noisy retrievals for DOAS algorithms. Is there screening of negative values that could produce a positive bias?

How is the varying coverage at the highest latitudes during a month reported? For example, if a grid point only has values for the last 20 days of the month, is there an associated average date to record this bias?

**For $NO_2$:**
The paper describes/cites two possible methods for determining the stratospheric overburden, and then concludes with a reference to a third paper for details. Which method is used to get the tropospheric estimates used here? Is there any information present in the Level 2 products for the tropospheric ozone for the masked regions where is it essentially assumed to be zero?  Can the cloudy scenes (say >90%) that do not have information on the tropospheric $NO_2$ be used to confirm the stratospheric column amounts, that is, recognize that there is only information on the above cloud $NO_2$ and use the appropriate AMFs? Are the total column $NO_2$ amounts recomputed after the redetermination of the tropospheric amounts or are the original estimates with stratospheric air mass factors used?

**For BrO:**
The use of an equatorial offset would seem to mean that the Level 3 cannot provide independent trends for 5°S to 5°N. Do monthly maps confirm this? That is, do they just show the stable assumed BrO levels?

**For $SO_2$:**
In Figure 7 and the following text, it is noted that Metop-C retrievals are much less noisy at high SZA and that they are 0.5 DU lower in general. Is this "in general" for a typical measurement or is it for a global average due to the high $SO_2$ values at high SZA for Metop-A & -B? The color scale is not ideal but I do not see a 0.5 DU difference in the tropics.  While the DOAS retrieval for Metop-C is different from the Metop-A & -B one (312 nm versus 315 nm and AMF at 313 nm versus 320 nm), it is not clear how this leads to the much poorer performance of the latter products at high SZA. Are the signal levels of the three instruments much different? (Perhaps some quantitative information on the degradation over time of the signals for the three sensors would be explanatory for some of the features of Figures 6 & 7.) Is it increased noise or bias, that is, I do not see negative values for the $SO_2$ at high SZA, just large positive ones? Also, what is the source of the higher values for Metop-A in 2019?

Figure 23 shows many negative values for $SO_2$ but they are not in the scatter plots in Figure 22. Please explain. (This may also be the case for HCHO in Figure 20.) Is any adjustment made to the total least squares regression for the differences in the precision for the satellite data values versus the Pandora data values?

Do major volcanic eruptions create perturbations in the monthly zonal mean products? The color scales chosen for the figures do not reveal any.

**Minor edits / Technical corrections**
Page 2, L17 irradance → irradiance
Page 2, L28 user friendly → user-friendly
Page 2, L18 Change to
  *The retrieval of trace gas columns from level 1B data includes spectral retrieval of slant columns of a trace gas and subsequently conversion to vertical columns.*
Page 2, L31: includes → include
Page 3, L4: is → are
Page 3, L18: side ways → sideways
Page 4, L14: range.In → range. In
Page 4, L16: $NO_2$ → An $NO_2$
Page 4, L19: air mass factor → air mass factors (AMFs)
Page 4, L19/20: Change to
  *Vertical distribution profiles are essential a priori information used in the calculation of AMFs.*
Page 4, Line 21/22: Change to
  *Based on the initial result of the ozone vertical column retrieval,*
Page 5, Line 4: intra cloud → intra-cloud
Page 5, Line25/25: Change to
  *The initial total VCD is retrieved assuming an unpolluted troposphere. Therefore, the air mass factor is weighted toward to stratospheric $NO_2$, whereas the tropospheric $NO_2$ amount is assumed to be negligible.*
Page 6, Line 25: Change to
  *The DOAS fit for water vapour retrievals takes into account $O_2$ and $O_4$ cross-sections ...*
Page 9, Line 21: $So_2$ → $SO_2$
Page 9, Line 24: a.s.l. → above sea level
Page 13, Line 7: low quality → low-quality
Page 13, Figure 1: overlayed → overlaid
Page 14, Line 25: data is → data are (twice)
Page 16, Line 11: Change to
  *Despite the fact that large numbers of observations are ...*
Page 18, Line 19: low quality → low-quality
Page 18, Line 22: Change to
  *The noise levels of monthly GOME-2A data are significantly higher than those of GOME-2B and C. This is mainly related to less ...*
Page 23, Line 13: tropopsheric → tropospheric
Page 29, Line 11: is higher → are higher
Page 35, Line 3: studies shows → studies show & when MAX-DOAS → when the MAX-DOAS
Page 37, Figure 22: Vertical Axis Label -- Occurancy → Occurrence
Page 45, L4: S. ichi Kurokawa → S. Kurokawa

---

## Author Response (AR1)

We thank the reviewers for their useful comments. We understand that these comments are mostly positive while minor corrections/revisions are necessary. We have addressed the reviewer's comments on a point-to-point basis as below for consideration. All page and line numbers refer to the marked-up version of the manuscript.

Chan et al. presented a new suite of L3 GOME-2 product including total O3, total and tropospheric NO2, total H2O, BrO, HCHO, and SO2. They inter-compared data from 3 sensors and evaluated the dataset with ground-based observations. The gridding strategy for producing L3 data is also documented. A user-friendly and well-documented L3 product presented is extremely valuable for the community and for wider usership. The paper is clearly written but additional information that guides a potential user may be added to improve the manuscript. I recommend the paper and data to be published in ESSD if the following comments are addressed.

Main comments

As the intended users for the L3 data include non-expert users, it would be useful for authors to provide advices or guidelines (including recommendations and/or cautions), but this is in general inadequate in the current manuscript. For example, do you recommend to use monthly average or daily data for a specific species, based on the evaluation? Is the current evaluation adeqaute the characterize the error statistics?

Ans: It is not very easy to give general recommendation as different users are using the data for different applications. For example, some users are using level 3 products for the investigation of long-term variability (Eleftheratos et al., 2019), in this case it would be reasonable to use monthly data. While someone uses the data for the monitoring/visualization of ozone hole over Antarctic (TOP STORY: Ozone hole above Antarctica, https://acsaf.org/), which is making use of daily data. We have provided information on the measurement error and variability (as standard deviation, see section 4), and based on these information the users should be able to decide using the data in which specific region at which specific time frame.

In addition, we have added notices for certain products, i.e., BrO and SO2, which are only validated at one location and might need to be further validated depending on the applications (page 38, line 15-16; page 42, line 7-9, Table 3).

Reference:

Eleftheratos, K., Zerefos, C. S., Balis, D. S., Koukouli, M.-E., Kapsomenakis, J., Loyola, D. G., Valks, P., Coldewey-Egbers, M., Lerot, C., Frith, S. M., Haslerud, A. S., Isaksen, I. S. A., and Hassinen, S.: The use of QBO, ENSO, and NAO perturbations in the evaluation of GOME-2 MetOp A total ozone measurements, Atmos. Meas. Tech., 12, 987–1011, https://doi.org/10.5194/amt-12-987-2019, 2019.

Performance statistics are important information for a user. However, very different amount of ground-based data are used for evaluation. Some are from global network, and some other are from only one site (e.g., Mexico city for SO2). I'd suggest to comment on how representative the derived bias & correlation are, and make proper recommendations to authors about the uncertainty.

Ans: We have added notice that for certain products, i.e., BrO and SO2, which are only validated at one location and might need to be further validated depending on the applications (page 38, line 15-16; page 42, line 7-9; Table 3).

Specific comments:

Page 5 Line 30: Unclear whethere the stratosphere-troposphere separation is done on the initial total vertical columns or total slant columns

Ans: Stratospheric $NO_2$ columns are derived from the initial total vertical columns using a spatial filtering approach (Wenig et al., 2004, Valks et al., 2011). This method masks out potentially polluted areas and then applies a low-pass filter in the zonal direction to derive the stratospheric component. The tropospheric $NO_2$ vertical columns are then derived from the residual tropospheric slant columns using a more accurate tropospheric air mass factor which considers the effects of clouds and $NO_2$ profile shape from a chemistry transport model. Finally, the derived tropospheric columns are used to correct the initial total $NO_2$ column under polluted conditions to provide an estimate of the total vertical column. We have revised the description in the revised version (page 6, line 3-13).

Page 12 Line 29: Here observation at Mexico City is used for evaluation, but the above L2 description stated that retrieval algorithm assumes SO2 from volcano. What is the implication of this inconsistency?

Ans: The level 2 product includes SO2 columns calculated with different scenario, i.e., volcanic emissions (e.g., eruptions and degassing) and anthropogenic emissions. For volcanic emissions, the AMFs are calculated assuming the SO2 plume follows a Gaussian profile shape with central plume height at 15.0km, 6.0km and 2.5km above sea level. For emission from anthropogenic sources, the AMFs are calculated assuming a homogeneous layer from surface up to 1km. And the GOME-2 SO2 product typically refers to the one assuming plume height at 6km.

Volcanic SO2 is a very specific topic due to the differences in plume injection height for each eruption. And the validations are often done in case study bases which do not provide long term statistic. The major objective of this study is to evaluate the long-term statistic and stability of the level product. Therefore, we focus on area with significant sources spinning over longer period. For sure the inconsistency in the a priori profile assumption and emission type will have some impacts on the result. We have added this information in the new version (page 10, line 2-8, page 42, line 5-9).

Page 14 Eq (2) & Eq (3): I do not see any results related to Eq (2) and (3) presented in the manuscript. Are they included in the L3 product?

Ans: Error and standard deviation are included in the level 3 products. Examples of error and standard deviation are now added in section 4 (page 19, line 21 to page 22, line 2, Fig. 6-9).

Table 2 does not appear to have enough information. Can merge with Table 4

Ans: We have merged Table 2 in Table 4 (Table 3 in the revised version).

Section 5.1.1 & 5.1.2: Use separate paragraphs for each species. So readers can find information on species of interest easily.

Ans: We have separated each species in separated paragraphs.

Table 3: It would be helpful if you also list relative metrics for biases.

Ans: The relative terms are sometime confusing, for example, the SO2 columns are expected to be close to zero, and the relative term could be extremely high (up to few hundred percent) which does not really represent the accuracy of the dataset. Instead, we have added relative bias in the Table 4 for the ground-based comparison.

Page 25 Line 9. Discrepancy appears to be large also in Northern mid-latitude based on fig. 7.

Ans: We have now mentioned the discrepancy in Northern mid-latitude as well and added a sentence explaining this is likely related to the background correction process of different sensors (page 30, line 7-8).

Page 34 Line 7-8: does this indicate that a user should use monthly HCHO data for analysis, or daily data should be used with caution?

Ans: We have added a notice that mentions the daily data is noisy and would be useful to further the data spatially when looking into day-to-day variability or directly use the temporal averaged product, i.e., the monthly product (page 40, line 4-6).

Table 4: Are reported metrics derived from daily products or monthly products? Consider to report as accuracy (bias) and precision, as these are often useful metrics when model-satellite comparison is done.

Ans: We have added a remark at the bottom of the table mentioning what data is used to derive these numbers. We have also added relative terms in table 3.

Minor text edits:

Page 2 Line 12-13: "Together with its successors, Global Ozone Monitoring Experience 2 ...". I'd suggest to rephrase. It reads like GOME-2 and its successors (that would be GOME-3...).

Ans: We have revised the sentence (page 2, line 13-18).

Page 2 Line 28: In order "to"

Ans: Done.

Page 2 Line 31: includes -> include

Ans: Done.

Page 9 Line 21: capitalize O in So2

Ans: Done.

Page 21 Line 25; Page 22 Line 2; Page 23 Line 1; Page 29 Line 21 and elsewhere: "result" -> "result in"

Ans: Done.

Figure 7: Texts are too small to read. May consider to remove repeated color bars for 1st and 2nd columns and increase the font size of the 3rd column.

Ans: We have enlarged the label of the axis and removed the colorbar of 1$^{st}$ and 2$^{nd}$ column (Fig. 11 in the revised version).

The paper by Chan et al. presents the GOME-2 Level-3 data of total column ozone, total and tropospheric column nitrogen dioxide, total column water vapour, total column bromine oxide, total column formaldehyde and total column sulphur dioxide. The topic fits well to the aims and scopes of ESSD. The manuscript is overall very well-written and has a clear structure. The described methods and validation results seem reasonable. I would favorably recommend a publication after the revised manuscript could

Specific comments

In Introduction, it would be nicer to mention instruments like SCIAMACHY, OMI, and TROPOMI that measure atmospheric constituents within UV-VIS.

Ans: We have included SCIAMACHY, OMI, and TROPOMI in the introduction (page 2, line 13-18).

As you mentioned that cloud parameters done by the OCRA/ROCINN algorithms, the cloud model CRB (Clouds-as-Reflecting-Boundaries) was used for GOME-2 A/B/C. Why the cloud model CAL (Clouds-As-Layers) wasn't used since CAL has been included in OCRA/ROCINN and (Loyola et al., 2018)? A new surface albedo climatology based on hyperspectral UV-VIS measurements has been introduced for TROPOMI (Loyola et al., 2020). Would it be also applied to GOME-2 data processing as well? Please extend the discussion in the manuscript.

Ans: The latest version of the OCRA/ROCINN algorithm applied to TROPOMI includes the retrieval of cloud-top height and cloud optical thickness using the cloud model CAL (Clouds-As-Layers) (Loyola et al., 2018) and this new feature has been implemented to process TROPOMI data. The CAL model included in OCRA/ROCINN has also been implemented in our prototype GOME-2 NO2 algorithm, as described in Liu et al., 2020. The new NO2 algorithm also uses an improved directionally dependent Lambertian-equivalent reflectivity (DLER) for AMF calculation. It is planned to implement the prototype GOME-2 NO2 algorithm in a future version of the operational GDP processor. We have now mentioned this in the cloud product description section (page 10, line 34 to page 11, line 3).

Page 2, Line 13-15: Please rewrite "Together with its successors . . . 25 years".

Ans: We have rewritten the sentence (page 2, line 13-18).

Page 2, Line 25: Why "usually"? Any other format for expressing trace gases columns?

Ans: We have deleted "usually" from the sentence (page 2, line 28).

Page 4, Table 1: Would it be possible to summarize the major differences between GDP 4.8 and GDP 4.9, except for different sensors?

Ans: The GDP 4.9 was introduced for GOME-2C and includes updated instrument specific retrieval settings for $NO_2$ and $SO_2$. For $NO_2$, the alternative DOAS fitting-window 430.2-465 nm is used (because of calibration issues for GOME-2C for wavelengths < 430 nm). For $SO_2$, improved DOAS fitting settings are used, and the wavelength region has been changed to 312-325 nm (Valks et al., 2019). We have added this information in the table footnote (Table 3).

Page 21, Sect. 5.1.1: Different instrument characteristics like scan angle dependency and polarization sensitivity seem to have higher impact on UV species (O3, BrO, HCHO) than VIS species (NO2 and H2O). Is that true?

Ans: It is true, see Merlaud et al., 2020, Pinardi et al., 2020.

Reference:

Merlaud, A., Theys, N., Hendrick, F., van Gent, J., Pinardi, G., Van Roozendael, M., Chan, K. L., Heue, K. P., and Valks, P.: Validation report of GOME-2 GDP 4.9 BrO column data for MetOp-C Operational Readiness Review, Tech. rep., EUMETSAT AC SAF, https://acsaf.org/docs/vr/Validation_Report_OTO_BrO_May_2020.pdf, 2020.

Pinardi, G., Yu, H., Van Roozendael, M., Van Gent, J., Chan, K. L., and Valks, P.: O3M SAF Validation Report for GOME-2C Total HCHO, https://acsaf.org/docs/vr/Validation_Report_NTO_OTO_HCHO_May_2020.pdf, 2020b.

Page 23, Line 9: This sentence should be placed in the previous paragraph.

Ans: Done.

Page 23, Line 17: was observed by . . .

Ans: Done.

Page 25, Sect. 5.2.1: If available, could the authors provide relative differences for total ozone comparison between GOME-2 and ground-based measurements, which has been widely used in the total ozone validation?

Ans: We have now indicated the relative bias for ozone column comparison (Table 3).

Page 39, Sect. 6: I suggest to add a paragraph to discuss the usefulness of GOME-2 Level-3 data. As compared to TROPOMI, OMI, and other data, would the authors still recommend it to the community?

Ans: Compared to OMI and TROPOMI which are measuring at noon to early afternoon, GOME-2 measurements in the morning provides addition information on the temporal and diurnal variation of these atmospheric species. This feature makes the GOME-2 data record a unique product and therefore highly recommended to the community, especially for those focus on photochemistry and diurnal variations. We have added this information in the introduction and summary (page 3, line 3-5, page 43, line 3-6).

References

Loyola, D. G., Gimeno García, S., Lutz, R., Argyrouli, A., Romahn, F., Spurr, R. J. D., et al. (2018). The operational cloud retrieval algorithms from TROPOMI on board Sentinel-5 Precursor. Atmos. Meas. Tech. 11, 409–427. doi:10.5194/amt-11-409-2018

Loyola, D. G., Xu, J., Heue, K.-P., and Zimmer, W. (2020). Applying FP_ILM to the retrieval of geometry-dependent effective lambertian equivalent reflectivity (GE_LER) daily maps from UVN satellite measurements. Atmos. Meas. Tech. 13, 985–999. doi:10.5194/amt-13-985-2020

Review of Global Ozone Monitoring Experiment-2 (GOME-2) Daily and Monthly Level 3 Products of Atmospheric Trace Gas Columns by Ka Lok Chan et al.

This paper gives a full description of a new set of daily and monthly Level 3 products created from Level 2 Products (Total Column O3, Total and Troposphere NO2, Total H2O, Total BrO, total HCHO, and total SO2) estimated by using measurements from the Eumetsat GOME-2 instruments on the Metop platforms. It provides a clear description of the choices and statistical approaches used for the different species as well as validation versus ground-based measurements and the original Level 2 products. The paper includes summaries of the strengths and weaknesses for each component together with extensive citations of the literature.

There is always a loss of information in the creation of a Level 3 product in exchange for the filtering, weighting, and selection exercised by the developer. Is there a way to preserve or access the information on the a priori profiles and the viewing geometry for the Level 3 products? If not, then it becomes difficult to determine whether the column values are consistent with comparative profile measurements.

Ans: This is a very good suggestion. However, it is very difficult to provide additional information such as averaging kernel, a-priori vertical profile, viewing and solar geometries in the level 3 product, as these parameters cannot easily be expressed as an average. The objective of producing this new level 3 product is to provide a user-friendly data set for the broader community especially the non-satellite measurement experts. If the users are interested in that more detailed information, they are probably more advanced users and being able to use the level 2 data directly.

Given the differences in SZAs and local times is it worth separating out the ascending and descending measurements? Can the repeated measurements at high latitudes from ascending and descending portions of the orbits be used to investigate diurnal variations?

Ans: The reviewer is correct that the ascending and descending measurements in the polar regions can potentially be used to analyse diurnal variations. But there isn't any simple solution that can separate these measurements without creating additional data set. Therefore, for the simplicity, the level 3 product does not separate ascending and descending measurements.

It should be possible to identify and flag many of the Level 1 spectra affected by charged particle events within the South Atlantic Anomaly. Is this information available from the Level 2 Products? While the expected effects are an increase in the recorded signal, they will produce noisy retrievals for DOAS algorithms. Is there screening of negative values that could produce a positive bias?

Ans: The level 1 spectra quality will directly affect the DOAS fit result and leads to larger residual. The DOAS fit quality and uncertainties are provided in the L2 products, and a SAA flag is included as well. For the L3 products, we filter data based on the DOAS fit quality, solar geometry, and cloud cover. But did not filter negative value specifically as it is likely to result in a positive bias on average.

How is the varying coverage at the highest latitudes during a month reported? For example, if a grid point only has values for the last 20 days of the month, is there an associated average date to record this bias?

Ans: The product includes the number of observations, i.e., the grid cell has been covered how many times within a day or a monthly. We have added this information in the manuscript (page 15, line 1).

For NO2:

The paper describes/cites two possible methods for determining the stratospheric overburden, and then concludes with a reference to a third paper for details. Which method is used to get the tropospheric estimates used here? Is there any information present in the Level 2 products for the tropospheric ozone for the masked regions where is it essentially assumed to be zero? Can the cloudy scenes (say >90%) that do not have information on the tropospheric NO2 be used to confirm the stratospheric column amounts, that is, recognize that there is only information on the above cloud NO2 and use the appropriate AMFs? Are the total column NO2 amounts recomputed after the redetermination of the tropospheric amounts or are the original estimates with stratospheric air mass factors used?

Ans: The current operational GOME-2 $NO_2$ processor uses a spatial filtering approach to estimate the stratospheric burden, as described in Wenig et al., 2004 and Valks et al., 2011. This method masks out potentially polluted areas. An improved STS algorithm, STREAM, has been implemented in our prototype GOME-2 $NO_2$ algorithm (Liu et al. 2019). The STREAM algorithm uses measurements over clean, remote regions as well as over clouded scenes (where the tropospheric column is effectively shielded) via various weighting factors (Beirle et al., 2016). It is planned to implement the prototype GOME-2 $NO_2$ algorithm in a future version of the operational GDP processor. The derived tropospheric columns are used to correct the initial total NO2 column under polluted conditions to provide an estimate of total vertical column. We have updated the text in this section accordingly (page 6, line 3-13).

Reference:

Beirle, S., Hörmann, C., Jöckel, P., Liu, S., Penning de Vries, M., Pozzer, A., Sihler, H., Valks, P., and Wagner, T., The STRatospheric Estimation Algorithm from Mainz (STREAM): Estimating stratospheric $NO_2$ from nadir-viewing satellites by weighted convolution, Atmos. Meas. Tech. 9, 2753-2779, doi:10.5194/amt-9-2753-2016, 2016.

For BrO:

The use of an equatorial offset would seem to mean that the Level 3 cannot provide independent trends for 5°S to 5°N. Do monthly maps confirm this? That is, do they just show the stable assumed BrO levels?

Ans: The background correction is on the slant column level. The resulting vertical columns are largely dependent on the AMFs, which is mainly influenced by the climatological profile, viewing and solar geometries. As these factors show seasonal patterns, the resulting BrO columns also show a seasonal pattern.

For SO2:

In Figure 7 and the following text, it is noted that Metop-C retrievals are much less noisy at high SZA and that they are 0.5 DU lower in general. Is this "in general" for a typical measurement or is it for a global average due to the high SO2 values at high SZA for Metop-A & -B? The color scale is not ideal but I do not see a 0.5 DU difference in the tropics. While the DOAS retrieval for Metop-C is different from the Metop-A & -B one (312 nm versus 315 nm and AMF at 313 nm versus 320 nm), it is not clear how this leads to the much poorer performance of the latter products at high SZA. Are the signal levels of the three instruments much different? (Perhaps some quantitative information on the degradation over time of the signals for the three sensors would be explanatory for some of the features of Figures 6 & 7.) Is it increased noise or bias, that is, I do not see negative values for the SO2 at high SZA, just large positive ones? Also, what is the source of the higher values for Metop-A in 2019?

Ans: The 0.5 difference mentioned here refers to the global average. We have further clarified that in the revised manuscript (page 27, line 31-32). The colour scale is not idea for the visualization of this type of small differences. However, it is essential for the visualization of volcanic plume.

The SO2 absorption is stronger at shorter wavelengths in the UV, and therefore the optimized fitting window (312-325 nm) for GOME-2C leads to enhancement of signal to noise. The spectral fitting window is optimized based on several factors, such as, the interference from other species and instrument specific issues (stray light, polarization sensitivity, degradation including temporal changes in the instrument slit-function etc). Due to the difference in age of the instruments, as well as instrument specific calibration issues, the signal to noise can be very different among the three instruments. The degradation of the signal cannot easily be seen/quantified as there is a background correction procedure which corrected the degradation to some extent, and the aging/degradation of the GOME-2 instruments mainly affect the noise level. However, instrument specific calibration issues and changes in the instrument slit function can result in biases that cannot be fully resolved with the background correction. For the slightly higher SO2 in 2019 measured by GOME-2A, we have revisited the data and it is in general higher over the entire globe, presumably related to limitations in the background correction. We have now mentioned this issue in the manuscript (page 30, line 12-13).

Figure 23 shows many negative values for SO2 but they are not in the scatter plots in Figure 22. Please explain. (This may also be the case for HCHO in Figure 20.) Is any adjustment made to the total least squares regression for the differences in the precision for the satellite data values versus the Pandora data values?

Ans: They were just hidden below the axis. We did not apply any filtering or adjustment for the regression. To avoid confusion, we have also covered the negative part in the scatter plots (Fig. 26).

Do major volcanic eruptions create perturbations in the monthly zonal mean products? The color scales chosen for the figures do not reveal any.

Ans: Volcanic eruptions create perturbations in the monthly maps as can be seen in Fig. 5. For example, the enhancement of SO2 over Western Pacific Ocean is related to the eruption of volcano Taal in Philippines. The SO2 plume can also be seen in the daily map (Fig. 4). However, the intensity has been averaged temporally and longitudinally which makes it less significant to be visible in the monthly zonal plot.

Minor edits / Technical corrections

Page 2, L17 irradance → irradiance

Ans: Done

Page 2, L28 user friendly → user-friendly

Ans: Done

Page 2, L18 Change to

The retrieval of trace gas columns from level 1B data includes spectral retrieval of slant columns of a trace gas and subsequently conversion to vertical columns.

Ans: Done

Page 2, L31: includes → include

Ans: Done

Page 3, L4: is → are

Ans: Done

Page 3, L18: side ways → sideways

Ans: Done

Page 4, L14: range.In → range. In

Ans: Done

Page 4, L16: NO2 → An NO2

Ans: Done

Page 4, L19: air mass factor → air mass factors (AMFs)

Ans: Done

Page 4, L19/20: Change to

Vertical distribution profiles are essential a priori information used in the calculation of AMFs.
Ans: Done

Page 4, Line 21/22: Change to

Based on the initial result of the ozone vertical column retrieval,

Ans: Done

Page 5, Line 4: intra cloud → intra-cloud

Ans: Done

Page 5, Line25/25: Change to

The initial total VCD is retrieved assuming an unpolluted troposphere. Therefore, the air mass factor is weighted toward to stratospheric NO2, whereas the tropospheric NO2 amount is assumed to be negligible.

Ans: Done

Page 6, Line 25: Change to

The DOAS fit for water vapour retrievals takes into account O2 and O4 cross-sections …

Ans: Done

Page 9, Line 21: So2 → SO2

Ans: Done

Page 9, Line 24: a.s.l. → above sea level

Ans: Done

Page 13, Line 7: low quality → low-quality

Ans: Done

Page 13, Figure 1: overlayed → overlaid

Ans: Done

Page 14, Line 25: data is → data are (twice)

Ans: Done

Page 16, Line 11: Change to

Despite the fact that large numbers of observations are …

Ans: Done

Page 18, Line 19: low quality → low-quality

Ans: Done

Page 18, Line 22: Change to

The noise levels of monthly GOME-2A data are significantly higher than those of GOME-2B and C. This is mainly related to less …

Ans: Done

Page 23, Line 13: tropopsheric → tropospheric

Ans: Done

Page 29, Line 11: is higher → are higher

Ans: Done

Page 35, Line 3: studies shows → studies show & when MAX-DOAS → when the MAX-DOAS

Ans: Done

Page 37, Figure 22: Vertical Axis Label -- Occurancy → Occurrence

Ans: Done

Page 45, L4: S. ichi Kurokawa → S. Kurokawa

Ans: Done

"The scientific results and conclusions, as well as any views or opinions expressed herein, are those of the author and do not necessarily reflect those of NOAA or the Department of Commerce."

Ans: Done